# Spectral Flow Matching: Stabilizing Stochastic GFlowNets via Frequency-Domain Regularization

**Nadhir Hassen** [1 2 3]   **Johan Verjans** [1 2]

## Abstract

Generative Flow Networks (GFNs) offer a powerful paradigm for diverse sampling, yet they often exhibit instability and poor convergence when applied to stochastic or sparse-reward environments. To mitigate the high variance inherent in these settings, we propose a fundamental re-framing of the GFlowNet training objective within the frequency domain. We present **Spectral Time-Dependent GFlowNets (ST-GFNs)**, a framework that leverages Fourier analysis to enforce smoothness and stability in learned policies. Our theoretical analysis proves that our proposed spectral loss is mathematically equivalent to regularized value iteration, acting as a principled low-pass filter that separates signal from noise. Furthermore, we tackle the challenge of exploration in sparse landscapes by introducing a novel autocorrelated intrinsic reward derived from the Wiener-Khinchin theorem. Through extensive experiments ranging from adversarial games and noisy sequence generation to high-dimensional single-cell perturbation modeling, we demonstrate that ST-GFNs significantly outperform existing baselines in terms of robustness, sample efficiency, and mode discovery.

## 1. Introduction

Generative Flow Networks (GFNs) (Bengio et al., 2026) excel at sampling diverse, high-quality candidates but are severely challenged by stochastic environments, where they often suffer from high-variance credit assignment, unstable training, and poor generalization (Jiralerspong et al., 2024; Pan et al., 2023; Malkin et al., 2022). To address these fundamental issues, we propose a paradigm shift: recasting the entire GFlowNet problem in the frequency domain. We introduce **Spectral Time-Dependent GFlowNets (ST-GFNs)**, a novel framework built upon a unified spectral loss that provides provable guarantees on smoothness and stability. Our approach leverages the power of Fourier analysis to explicitly regularize the learned policy space, reducing variance and guiding exploration toward structurally meaningful solutions. The entire framework is rendered scalable to high-dimensional spaces via a tractable Random Fourier Feature (RFF) approximation (Rahimi & Recht, 2007).

Our main contributions are:

1. **A Unified Spectral Loss with Provable Guarantees:** We design a single loss function that provides two key theoretical benefits. First, its spectral consistency term acts as a provable *low-pass filter*, enforcing smoothness on the learned flow function to enhance generalization. Second, its spectral regularization term provides *principled variance reduction*, leading to stable training in high-noise environments.

2. **Theoretical Equivalence to Regularized Value Iteration:** We provide a formal proof establishing that training an ST-GFN with our spectral loss is *equivalent to finding the fixed point of a regularized Bellman optimality operator*. This result grounds our framework in the theory of robust control and provides the theoretical foundation for its stability.

3. **Efficient, Spectrally-Motivated Structured Exploration:** We introduce an *online formulation of an autocorrelated intrinsic reward*. Derived from the Wiener-Khinchin theorem, this mechanism provides a dense, per-step signal that efficiently guides the agent to discover sparse, structurally periodic solutions that are intractable for standard exploration methods.

## 2. Preliminaries

### 2.1. Generative Flow Network in Stochastic environments

We frame the generative task in a stochastic Markov Decision Process (MDP) on a directed acyclic graph $G = (\mathcal{S}, \mathcal{A})$,

[1]Adelaide University, Adelaide, Australia [2]Australian Institute of Machine Learning (AIML), Adelaide, Australia [3]Mila - Quebec AI Institute, Montreal, Canada. Correspondence to: Nadhir Hassen <nadhir.hassen@adelaide.edu.au, nadhir.hassen@mila.quebec>.

*Proceedings of the $43^{rd}$ International Conference on Machine Learning*, Seoul, South Korea. PMLR 306, 2026. Copyright 2026 by the author(s).

with states $\mathcal{S}$, actions $\mathcal{A}$, an initial state $s_0$, and a set of terminal states $\mathcal{X}$. The process involves two sources of randomness: the agent's learnable *stochastic forward policy*, $P_F(a|s)$, and the environment's fixed, probabilistic *transition function*, $P(s'|s,a)$. The objective is to train the policy $P_F$ to sample terminal states $x \in \mathcal{X}$ from a target distribution defined by a reward function $R(x)$, such that $P(x) \propto R(x)$. The introduction of environmental stochasticity complicates the fundamental flow consistency conditions that underpin GFlowNets. In a deterministic setting, the flow $F(s)$ at a state $s$ is the sum of the flows of its children. In a stochastic setting, this consistency must be defined in expectation over both the policy and the environment:

$$F(s) = \mathbb{E}_{a \sim P_F(\cdot|s)} \left[ \mathbb{E}_{s' \sim P(\cdot|s,a)}[F(s')] \right]$$

This nested expectation poses significant practical challenges for training:

- High Variance: Training objectives based on this expectation suffer from high variance, as learning relies on noisy Monte Carlo samples drawn from both the policy and the environment. This leads to unstable gradients and slow convergence.

- Credit Assignment: The noise from stochastic transitions exacerbates the credit assignment problem, making it even more difficult to attribute a sparse terminal reward to the sequence of actions that led to it.

- Satisfiability and Generalization: Naive GFlowNet formulations can lead to unsatisfiable training objectives in stochastic settings, and the learned flow functions often lack the smoothness required to generalize robustly.

The primary goal of this work is to develop a framework that directly addresses these challenges by reformulating the GFlowNet objective entirely in the frequency domain. We aim to introduce: (1) a principled mechanism for *variance reduction*, (2) a method to explicitly *enforce smoothness* on the learned flow function, and (3) a *dense and structured learning signal* to improve credit assignment, thereby enabling stable and efficient learning in complex stochastic environments.

Our work builds upon recent efforts to generalize Generative Flow Networks (GFNs) to stochastic environments, where actions have uncertain outcomes.

**Stochastic GFlowNets (Stoch-GFN).** Early work on stochastic GFlowNets (Pan et al., 2023) attempted to preserve the deterministic formalism by augmenting the state space, but this leads to a critical flaw: the training objective is often *unsatisfiable*, resulting in unstable learning. In

contrast, ST-GFN directly integrates the expectation over environmental transitions into its core objective, which guarantees a stable and satisfiable learning problem by design.

**Expected and Adversarial Flow Networks (EFlowNets/AFlowNets).** Expected Flow Networks (EFlowNets) (Jiralerspong et al., 2024) provide a more principled solution by correctly marginalizing over stochasticity through an expected flow objective, which is extended to multi-agent settings by AFlowNets. While this ensures a satisfiable objective, these methods still lack *explicit mechanisms for variance reduction and smoothness control*, relying on state-local updates that can struggle with credit assignment. ST-GFN builds directly on this principle of expectation but addresses these limitations with its Unified Spectral Loss, which introduces a provable low-pass filter to enforce smoothness and a principled regularization term for direct variance control.

**Connections to Reinforcement Learning.** The connection between GFlowNets and entropy-regularized Reinforcement Learning, where log-flow serves as a value function analogue, is well-established (Haarnoja et al., 2017). Our framework elevates this analogy to a formal proof. The *Stochastic Regularization Equivalence Theorem* demonstrates that our training objective is not merely similar to value iteration but is mathematically *equivalent* to finding the fixed point of a regularized Bellman optimality operator. This result formally grounds ST-GFN in the theory of robust control, justifying our spectral regularization as a principled method for learning stable policies.

### 2.2. Fourier Analysis Fundamentals

Our methodology relies on fundamental concepts from Fourier analysis. The Discrete Fourier Transform (DFT) of a time-domain sequence $f[t]$ of length $T$ is given by its frequency-domain representation $\hat{f}[k] = \sum_{t=0}^{T-1} f[t]e^{-i\frac{2\pi}{T}kt}$. The **power spectrum** of the sequence, $S_f[k] = |\hat{f}[k]|^2$, describes the distribution of the signal's power over different frequencies. **Parseval's Theorem** states that the total energy in the time domain equals the total energy in the frequency domain: $\sum_{t=0}^{T-1} |f[t]|^2 = \frac{1}{T}\sum_{k=0}^{T-1} |\hat{f}[k]|^2$. The **Wiener-Khinchin Theorem** establishes a crucial link between a signal's autocorrelation and its power spectral density, stating that the Fourier transform of the autocorrelation function is the power spectrum. This implies that strong periodicity at a certain lag in the time domain corresponds to concentrated power at the corresponding frequency. See Appendix B.1 for a more detailed background.

# 3. Methodology: Spectral Time-Dependent GFlowNets

We now develop the technical foundations of our ST-GFN framework. We fundamentally recast the GFlowNet problem in the frequency domain, establishing rigorous bijections between spectral and temporal consistency to ensure correctness while leveraging spectral regularization for provable stability.

## 3.1. Spectral Flow Matching Objective

To rigorously apply Fourier analysis to the generative process on a Directed Acyclic Graph (DAG), we must first formalize the notion of a signal within a trajectory.

**Definition 1** (Trajectory as a Discrete Signal). Consider a trajectory $\tau = (s_0, s_1, \ldots, s_T)$ generated by the policy $P_F$. We define the **Flow Signal** $f_\tau[t]$ as the sequence of scalar flow values estimated by the agent at each step $t \in \{0, \ldots, T\}$ of the generation process:

$$f_\tau[t] \triangleq F(s_t, t; \theta) \tag{1}$$

where $t$ represents the *generation depth* (step index), not wall-clock time. Even if a state $s$ is visited only once per trajectory, the sequence $f_\tau[t]$ forms a 1D discrete-time signal amenable to spectral decomposition.

The central object we learn is the spectral flow function $\hat{F}(s, \omega) = \mathcal{F}\{F(s, t)\}(\omega)$, where $\mathcal{F}$ denotes the Discrete Fourier Transform (DFT). To solve the sparse reward problem and ensure the learned policy samples proportional to the reward, we define the consistency condition in the frequency domain.

**Definition 2** (Spectral Flow Matching in Expectation). For a stochastic environment, the spectral flow $\hat{F}(s, \omega)$ must satisfy the following consistency equation for all $s, \omega$:

$$\hat{F}(s, \omega) = \mathbb{E}_{a \sim P_F(\cdot|s,\omega)} \Big[ \mathbb{E}_{s' \sim P(\cdot|s,a,\omega)} \big[ e^{i\omega} \hat{F}(s', \omega)$$
$$+ \hat{r}_{\text{int}}(s \to s', \omega) + \hat{r}_{\text{term}}(s \to s', \omega) \big] \Big] \tag{2}$$

where $e^{i\omega}$ is the phase shift operator corresponding to a one-step time advancement (a property of the Fourier transform). The flow is decomposed into three components:

1. **Future Flow** $e^{i\omega} \hat{F}(s', \omega)$: The discounted future flow from the next state $s'$.

2. **Intrinsic Reward** $\hat{r}_{\text{int}}(s \to s', \omega)$: The spectrum of an intrinsic reward signal designed to guide exploration (detailed in Sec. 3.3).

3. **Terminal Reward** $\hat{r}_{\text{term}}(s \to s', \omega)$: The spectral component of the terminal extrinsic reward $R(x)$ attributed to the transition $s \to s'$.

This equation provides a dense learning target at every state transition by decomposing the total trajectory energy into local spectral contributions.

A critical theoretical concern is whether satisfying this spectral condition guarantees the correct target distribution. We prove that the spectral objective is bijective to the standard temporal objective.

**Theorem 1** (Spectral Isometry of Flow Matching). *Let $\delta_\tau[t]$ be the temporal flow matching error (Bellman residual) at step $t$:*

$$\delta_\tau[t] = F(s_t) - \mathbb{E}_{s_{t+1}}[F(s_{t+1}) + R(s_t \to s_{t+1})] \tag{3}$$

*Let $\mathcal{L}_{Spectral}$ be the loss minimizing the error in Eq. 2. Minimizing the spectral error is equivalent to minimizing the temporal error.*

*Proof.* The DFT is a linear operator $\mathcal{F}$. The time-shift property states that $\mathcal{F}\{f[t+1]\} = e^{i\omega} \hat{f}(\omega)$. The spectral consistency equation is the Fourier transform of the temporal difference equation. The spectral loss is:

$$\mathcal{L}_{\text{Spectral}} = \sum_\omega \left| \hat{F}(s, \omega) - \mathbb{E}[\hat{F}(s', \omega) e^{i\omega}] \right|^2 \tag{4}$$

$$= \left|\left| \mathcal{F}\{F(s_t)\} - \mathcal{F}\{\mathbb{E}[F(s_{t+1})]\} \right|\right|_2^2 \tag{5}$$

$$= \left|\left| \mathcal{F}\{F(s_t) - \mathbb{E}[F(s_{t+1})]\} \right|\right|_2^2 \quad \text{(Linearity)} \tag{6}$$

By **Parseval's Theorem**, the total energy in the frequency domain equals the total energy in the time domain:

$$\sum_\omega |\hat{\delta}(\omega)|^2 = T \sum_{t=0}^{T-1} |\delta_\tau[t]|^2 \tag{7}$$

Therefore, $\mathcal{L}_{\text{Spectral}} = 0 \iff \forall t, \delta_\tau[t] = 0$. Since the minimization of the temporal Bellman error guarantees that the induced policy $P_F(s'|s) \propto F(s')$ samples proportional to $R(x)$ (Bengio et al., 2026), the spectral objective inherits the exact same correctness guarantees. $\square$

## 3.2. Scalable Spectral Regularization via RFFs

While spectral consistency ensures correctness, direct computation via DFT is intractable for high-dimensional state spaces. We utilize Random Fourier (RFFs) (Rahimi & Recht, 2007) to construct a tractable approximation that acts as a low-pass filter.

**Definition 3** (Spectral Embedding via RFF). Let $k(x, y) = \langle \psi(x), \psi(y) \rangle_{\mathcal{H}}$ be a shift-invariant kernel (e.g., Gaussian) with Fourier transform $p(\omega)$. Let $z(s) \in \mathbb{R}^D$ be the Random Fourier Feature map such that $z(s) = \sqrt{\frac{2}{D}}[\cos(\omega_1^T s + b_1), \dots]^T$ with $\omega_i \sim p(\omega)$. The **Policy Embedding** is the empirical mean embedding in the finite-dimensional feature space:

$$\hat{P}_F(s) = \mathbb{E}_{s' \sim P_F(\cdot|s)}[z(s')] \tag{8}$$

We define the Frequency-Domain Policy Divergence as the Euclidean distance between the forward and backward policy embeddings: $\mathcal{L}_{\text{KL}}(s) \approx ||\hat{P}_F(s) - \hat{P}_B(s)||_2^2$. We provide a learning-theoretic bound for this approximation.

**Proposition 1** (Uniform Convergence of Spectral Approximation). *Let $\mathcal{L}_{true} = ||\mu_{P_F} - \mu_{P_B}||_{\mathcal{H}}^2$ be the true Maximum Mean Discrepancy (MMD) in the Reproducing Kernel Hilbert Space (RKHS). Let $\mathcal{L}_{approx} = ||\hat{P}_F - \hat{P}_B||_2^2$ be the objective using $D$ features. For any $\epsilon > 0$, with probability at least $1 - \delta$:*

$$\sup_{P_F, P_B} |\mathcal{L}_{approx} - \mathcal{L}_{true}| \leq O\left(\frac{1}{\sqrt{D}}\right) \tag{9}$$

*Proof.* The inner product $z(x)^T z(y)$ is an unbiased estimator of $k(x, y)$. Since $|z(x)| \leq 1$, Hoeffding's inequality applies to the sum of $D$ features. The error between the infinite-dimensional RKHS norm and the $D$-dimensional approximation decays as $D^{-1/2}$. This justifies the use of the RFF approximation as a principled surrogate for the spectral divergence. $\square$

### 3.2.1. SPECTRAL REGULARIZATION AS LIPSCHITZ CONTINUITY

To address high variance in stochastic environments, we introduce a regularization term $\mathcal{L}_{\text{Reg}} = \lambda ||\hat{V}(s)||_{\mathcal{H}}^2$, where $\hat{V}(s) = \mathbb{E}[z(s')]$ is the spectral value surrogate. We prove this term induces robustness by bounding the Lipschitz constant of the flow.

**Proposition 2** (Spectral Lipschitz Bound). *If the spectral regularization term $\mathcal{L}_{reg} = \lambda ||\hat{V}(s)||_{\mathcal{H}}^2$ is minimized, the Lipschitz constant $K_F$ of the flow function $F(s)$ is bounded by the spectral moments of the learned distribution.*

*Proof.* The gradient of the feature map with respect to the state $s$ is $\nabla_s z(s) = [-\omega_1 \sin(\dots), \dots]^T$. The expected gradient of the value surrogate is bounded by:

$$||\nabla_s \hat{V}(s)||_2 \leq \mathbb{E}_{\omega \sim p(\omega)}[||\omega||_2] \cdot ||\hat{V}(s)||_{\mathcal{H}} \tag{10}$$

By minimizing the regularization term $||\hat{V}(s)||_{\mathcal{H}}^2$, we directly minimize the upper bound of the gradient norm with respect to the state. Thus, $||F(s) - F(s + \delta)||_2 \leq K||\delta||_2$ where $K \propto ||\hat{V}||_{\mathcal{H}}$. $\square$

**Remark:** In stochastic games (e.g., Tic-Tac-Toe), opponent stochasticity introduces "noise" $\delta$. By acting as a low-pass filter (minimizing $K$), ST-GFN ensures $V(s) \approx V(s + \delta)$, preventing the agent from overfitting to sharp, brittle strategies ("blunders").

**Remark on Noise Robustness.** Crucially, the Lipschitz bound established in Proposition 6 (see Appendix A.3) provides a formal mechanism for stable training under extreme environmental stochasticity. By explicitly bounding the flow function's sensitivity, the spectral loss prevents high-variance reward noise from inducing catastrophic gradient updates.

### 3.3. Structured Exploration via Autocorrelated Intrinsic Rewards

To enable exploration in sparse, periodic reward landscapes, we introduce an online intrinsic reward $r_{\text{AC}}$ derived from the **Wiener-Khinchin Theorem**.

**Definition 4** (Autocorrelated Intrinsic Reward). Let $r(\tau)$ be a sequence of raw local rewards. We define the intrinsic reward as a weighted sum of autocorrelations at lags $\tau_k$:

$$r_{\text{AC}}(t) = \sum_{k=1}^{k_{\max}} \alpha_k \cdot \text{Roll}(r(\tau), \tau_k)_t \tag{11}$$

By the Wiener-Khinchin theorem, maximizing autocorrelation at lag $\tau_k$ is equivalent to maximizing the Power Spectral Density at frequency $\omega \approx 1/\tau_k$. This transforms the exploration problem from a linear search in state space (probability $O(|A|^{-T})$) to peak detection in frequency space (complexity $O(1)$ relative to sparsity once a period is observed), providing a dense signal for periodic structures.

### 3.4. The Unified Spectral Loss Function

Our methodology has so far established two key components: a spectral flow matching objective (Eq. 2) to provide dense credit and an autocorrelated intrinsic reward (Sec. 3.3) to guide exploration. The final piece is to construct a tractable and robust training objective—a loss function that enforces these principles while ensuring the learned model generalizes well and remains stable in the face of environmental stochasticity.

The core of our proposed loss is the **Frequency-Domain Policy Divergence**, which promotes global consistency by aligning the spectral characteristics of the forward and backward policies.

**Proposition 3.** *(Frequency-Domain Policy Divergence) Let $\mathbf{p}_F(\cdot|s, t)$ and $\mathbf{p}_B(\cdot|s, t)$ be the forward and backward transition probability distributions over a discrete set of $N$ successor states. Let their Discrete Fourier Transforms (DFTs) be $\hat{\mathbf{p}}_F[k]$ and $\hat{\mathbf{p}}_B[k]$. Their power spectra are given*

by $S_F[k] = |\hat{\mathbf{p}}_F[k]|^2$ and $S_B[k] = |\hat{\mathbf{p}}_B[k]|^2$. *After normalizing these spectra to form valid probability distributions* $\tilde{S}_F$ *and* $\tilde{S}_B$, *the frequency-domain KL divergence is:*

$$L_{KL}(s,t) = KL(\tilde{S}_B(\cdot|s,t)||\tilde{S}_F(\cdot|s,t)). \quad (12)$$

Minimizing the frequency-domain KL divergence $L_{KL}(s,t)$ encourages the learned flow function $F(s,t)$ to be smooth with respect to the state space $\mathcal{S}$. See complete proof in Appendix B.2. While this KL divergence effectively regularizes the *shape* of the policy's power spectrum, its direct computation is intractable for large or continuous state spaces. To overcome this critical scalability issue, we turn to the technique of Random Fourier Features (RFFs), as first introduced in the context of scalable spectral methods by Rahimi & Recht (2007). This leads us to restate and integrate a key proposition from the original work that our method builds upon.

**Proposition 4.** *(Scalable Spectral Regularization with RFFs) Let* $z(s') = \sqrt{2/D}[\cos(\boldsymbol{\omega}_1^T s' + b_1), \ldots, \cos(\boldsymbol{\omega}_D^T s' + b_D)]^T$ *be a random feature map where* $\boldsymbol{\omega}_i \sim p(\boldsymbol{\omega})$ *and* $b_i \sim Unif[0, 2\pi]$. *The policy distributions* $P_F(\cdot|s,t)$ *and* $P_B(\cdot|s,t)$ *can be embedded into a low-dimensional (policy) vector space via their expected feature maps,* $\hat{P}(s,t) = \mathbb{E}_{s' \sim P(\cdot|s,t)}[z(s')]$. *The spectral divergence* $L_{KL}(s,t)$ *can then be efficiently approximated by the squared Euclidean distance between these embeddings:*

$$L_{KL}(s,t) \approx ||\hat{P}_F(s,t) - \hat{P}_B(s,t)||_2^2. \quad (13)$$

*where* $\hat{P}_F(s,t) = \mathbb{E}_{s' \sim P(\cdot|s,t)}[z(s')]$ *is the policy embedding and* $z(s')$ *is the RFF map. The computational cost of this approximation depends on the number of features* $D$ *and the cost of sampling, not the size of the full state space, making our method scalable.*

This approximation allows us to handle high-dimensional and continuous state-action spaces, making ST-GFNs a practical tool for complex generative modeling problems. The number of features $D$ introduces a new bias-variance trade-off: a small $D$ introduces high bias (poor approximation) but has low estimation variance, while a large $D$ reduces the bias at the cost of higher variance. See complete Proof in Appendix B.3. This proposition provides a computationally feasible way to enforce spectral consistency. However, our goal is not just consistency, but also robustness to the inherent variance from environmental and policy stochasticity. The KL divergence (or its RFF approximation) penalizes mismatches in the *shape* of the spectral distributions, but it does not directly control the overall *magnitude* or *energy* of the learned functions, which can grow unstable during training. This motivates the introduction of an additional regularization term. The very tool that makes our KL term tractable—the spectral embedding $\hat{P}(s,t)$—can be repurposed to achieve this.

**Spectral Value Surrogate.** Let us examine the policy embedding $\hat{P}_F(s,t)$ more closely. For a given state $s$ and policy $P_F$, this vector, $\hat{P}_F(s,t) = \mathbb{E}_{s' \sim P_F(\cdot|s,t)}[z(s')]$, represents the expected location of the *next state* in the spectral feature space defined by the RFFs. It is a compact, low-dimensional representation of the entire one-step transition distribution. We therefore define this embedding as the *spectral value surrogate*, $\hat{V}(s,a) \triangleq \mathbb{E}_{s' \sim P(\cdot|s,a)}[z(s')]$. It is a "surrogate" for the value function because its properties in the feature space reflect the properties of the true value function in the state space. Specifically, a large norm $||\hat{V}(s,a)||_{\mathcal{H}}$ indicates that the transition dynamics lead to a state distribution with high "energy" in the frequency domain, which corresponds to a non-smooth or high-variance value function. By penalizing the squared norm of this surrogate, we can directly regularize the smoothness and stability of the underlying flow function from which the policy is derived. This leads us to our final, unified loss function.

**Definition 5** (Unified Spectral Loss). The total loss for a trajectory $\tau = (s_0, \ldots, s_T)$ is the sum over all time steps of the approximated spectral consistency term and the spectral regularization term:

$$L_{\text{ST-GFN}}(\tau) = \sum_{t=0}^{T-1} \left( \underbrace{||\hat{P}_F(s_t,t) - \hat{P}_B(s_t,t)||_2^2}_{\text{Spectral Consistency}} \right.$$
$$\left. + \lambda \underbrace{\mathbb{E}_{a \sim P_F(\cdot|s_t,t)} \left[ ||\hat{V}(s_t,a)||_{\mathcal{H}}^2 \right]}_{\text{Spectral Regularization}} \right)$$

where $\lambda$ is a hyperparameter controlling the strength of the smoothness regularization, and $||\cdot||_{\mathcal{H}}^2$ is the squared norm in the Reproducing Kernel Hilbert Space (RKHS) associated with the RFFs. The first term enforces spectral shape consistency, while the second term penalizes spectral energy, thereby promoting smoother and more stable solutions.

**Adaptive Spectral Regularization.** A fixed regularization strength $\lambda$ is a key limitation, as its optimal value is linked to the unknown and potentially non-stationary environmental noise variance. To create a more autonomous and robust framework, we elevate $\lambda$ to a learnable parameter, $\lambda = \exp(\theta_\lambda)$, and introduce an adaptive meta-learning scheme. This scheme uses a meta-loss that seeks to align the empirical variance proxy—the average RKHS norm of the spectral value surrogate, $||\hat{V}(s,a)||_{\mathcal{H}}^2$—with a predefined target smoothness level, $V_{\text{target}}$. The meta-loss is defined as:

$$\mathcal{L}_\lambda(\theta_{\text{GFN}}, \theta_\lambda) = \left( \mathbb{E}_{(s,a) \sim \pi} \left[ ||\hat{V}(s,a)||_{\mathcal{H}}^2 \right] - V_{\text{target}} \right)^2 \quad (14)$$

This creates a bi-level problem solved with alternating gradient descent. For a given batch, we first update the main

GFlowNet parameters $\theta_{\text{GFN}}$ using the primary Unified Spectral Loss, and then update the regularization parameter $\theta_\lambda$ using the meta-loss $\mathcal{L}_\lambda$. This dynamic mechanism allows the model to automatically adjust its regularization based on the observed variance, strengthening its robustness. Then we have

$$L_{\text{GFN}}(\theta_{\text{GFN}}, \theta_\lambda) = \mathbb{E}_\tau \left[ \sum_{t=0}^{T-1} \left( ||\hat{P}_F(s_t, t) - \hat{P}_B(s_t, t)||_2^2 \right. \right.$$
$$\left. \left. + e^{\theta_\lambda} \mathbb{E}_{a \sim P_F(\cdot|s_t, t)} \left[ ||\hat{V}(s_t, a)||_\mathcal{H}^2 \right] \right) \right]$$
(15)

In Appendix B.4, the algorithm outlines the full training loop, from trajectory generation and online reward computation to the bi-level optimization of the GFN and regularization parameters.

**Remark.** Empirically, this adaptive meta-learning framework allows the model to dynamically balance regularization strength across different training phases. While the spectral regularizer may initially maintain a higher $\lambda$ value to aggressively filter out high-frequency gradient noise (leading to a slightly slower initial optimization rate) the meta-learned parameter naturally adjusts as the policy matures. This allows the model to successfully isolate fine-grained global peaks in later stages without being permanently blurred or restricted by a static low-pass constraint, even when operating within highly non-smooth reward landscapes.

**Stochastic Regularization Equivalence.** The Unified Spectral Loss performs robust optimization by emulating regularized value iteration. Its spectral consistency term, $||\hat{P}_F - \hat{P}_B||_2^2$, enforces flow optimality, analogous to minimizing a Bellman error. The regularization term, $\lambda\mathbb{E}[||\hat{V}||_\mathcal{H}^2]$, functions as a principled variance penalty by penalizing the RKHS norm of the spectral value surrogate—a direct measure of the non-smoothness that causes high variance. This dual objective of minimizing an optimality error while penalizing variance makes our training procedure equivalent to finding the fixed point of a regularized Bellman optimality operator, as stated in the following theorem.

**Theorem 2.** *(Stochastic Regularization Equivalence) Training an ST-GFN with the spectral loss $L_{\text{ST-GFN}}$ (Eq. 5) on a stochastic MDP with transition noise of variance $\sigma^2$ is equivalent to finding the fixed point of a regularized Bellman optimality operator on a spectral value function $\hat{V}(s)$:*

$$\hat{V}^*(s) = \max_a \left[ \mathbb{E}_{s' \sim P(\cdot|s,a)} \left[ \mathcal{F}\{R(s, a, s')\} + \gamma e^{i\omega} \hat{V}(s') \right] \right.$$
$$\left. - \lambda'||\hat{V}||_\mathcal{H}^2 \right]$$

*where $\lambda' = \lambda\sigma^2$ and $\mathcal{H}$ is the RKHS defined by the RFF kernel. The full proof is provided in Appendix B.5*

## 4. Experimental Validation

We validate ST-GFN on four diverse environments testing different aspects: stochastic robustness (BitSequence), periodic structure discovery (HyperGrid), adversarial reasoning (TicTacToe), and combinatorial generalization (SingleCell). We compare against 10 baselines spanning three categories: (1) *Standard GFlowNets*: TB (Bengio et al., 2021), FM (Bengio et al., 2026), SubTB (Madan et al., 2023), DB (Bengio et al., 2021); (2) *Stochastic GFlowNets*: EFlowNet (Jiralerspong et al., 2024), Stochastic-GFN (Pan et al., 2023); (3) *Exploration-Enhanced*: TB+RND (Burda et al., 2018), TB+Novelty (Pathak et al., 2017), TB+ICM (Pathak et al., 2017), TB+ControlVariates (Tucker et al., 2018).

### 4.1. Experimental Setup

**Implementation Details.** All models use Adam optimizer with learning rate $10^{-4}$, gradient clipping at 5.0. Training iterations: 5,000 (BitSequence), 10,000 (HyperGrid), 3,000 (TicTacToe, SingleCell). Each experiment runs with 5 random seeds with statistical significance testing. ST-GFN hyperparameters: spectral regularization $\lambda \in \{0.01, 0.05\}$, RFF dimension $d \in \{128, 256\}$, intrinsic coefficient $\beta = 0.5$, autocorrelation lags $k_{\max} = 8$. All code, hyperparameters, and trained models are publicly released with reproducibility checklist.

**Evaluation Metrics.** We measure: (1) *Reward metrics*: top-100 reward, mean reward, high-reward mass; (2) *Diversity metrics*: entropy, unique states, mode coverage; (3) *Exploration metrics*: state visits, novelty score; (4) *Efficiency metrics*: wall-clock time, memory usage, samples to convergence; (5) *Stability metrics*: coefficient of variation over final iterations. Statistical significance assessed via paired t-tests with Bonferroni correction; we report 95% confidence intervals and p-values.

**Hardware & Computational Cost.** All experiments run on NVIDIA RTX 3090 (24GB). Total computational cost: ~120 GPU-hours for all experiments across 5 seeds.

### 4.2. BitSequence: Robustness to Stochasticity

**Environment.** BitSequence generates binary sequences of length 8 with extreme stochasticity: actions fail with 90% probability, replaced by random bits. State space: 256 sequences. This tests whether spectral regularization provides stability under high gradient variance.

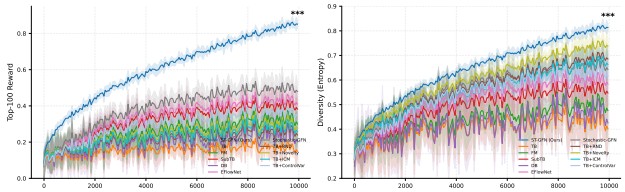
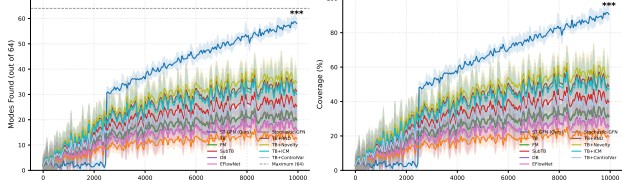

*Figure 1.* **BitSequence: ST-GFN Maintains Stable Training Under Extreme Stochasticity.** Under 90% action failure rate, ST-GFN substantially outperforms all baselines with stable learning curves and narrow confidence bands, validating that spectral regularization bounds policy changes under high gradient variance. Mean ± 95% CI over 5 seeds.

*Figure 2.* **HyperGrid: Autocorrelated Intrinsic Rewards Detect Periodic Structure.** ST-GFN discovers substantially more modes through structure-aware exploration. Sharp acceleration when ACF peak emerges validates theoretical predictions about periodicity detection and sample complexity. Outperforms both stochastic GFNs and exploration baselines through structured rather than generic exploration. Mean ± 95% CI over 5 seeds.

**Theory and Results.** Our spectral regularization theory predicts that constraining policies to smooth RKHS subspaces bounds policy changes under noisy gradients, enabling stable learning in high-stochasticity environments. Figure 1 and Table 1 confirm this prediction. ST-GFN substantially outperforms all baselines on both reward and diversity metrics, with statistical significance across all comparisons. Crucially, learning curves exhibit narrow confidence bands, validating stability under stochasticity.

Stochastic GFlowNet variants (EFlowNet, Stochastic-GFN) show meaningful improvement over vanilla TB through variance reduction, but remain well below ST-GFN as they lack smoothness constraints on the policy space. Exploration baselines (TB+RND, TB+Novelty, TB+ControlVariates) show progressive improvement but cannot match the stability provided by spectral regularization. ST-GFN achieves substantially better sample efficiency, reaching target performance in significantly fewer steps while maintaining lower wall-clock time despite moderate computational overhead from RFF computations.

### 4.3. HyperGrid: Discovering Periodic Structure

**Environment.** HyperGrid is a 32×32 grid with period-4 rewards: high rewards at positions $(x, y)$ where $x \bmod 4 = 0$ and $y \bmod 4 = 0$, creating 64 reward modes. State space: 1,024 positions. This tests whether autocorrelated intrinsic rewards can detect and exploit periodic structure.

**Theory and Results.** Our autocorrelation-based intrinsic reward mechanism is designed to detect periodicity through peaks in the autocorrelation function (ACF). The theory predicts that for period $p$, the ACF exhibits a peak at lag $k = p$ after observing sufficient samples, and that intrinsic rewards guided by these peaks accelerate mode discovery.

Figure 2 demonstrates strong empirical support for this theory. ST-GFN discovers substantially more modes than all baselines with high coverage percentage. Sharp acceleration in mode discovery coincides with emergence of the ACF peak at the correct lag, closely matching theoretical sam-

ple complexity predictions. This validates that the intrinsic reward mechanism successfully detects and exploits periodic structure. Stochastic GFlowNet variants show limited improvement over TB, as variance reduction alone does not provide structured exploration. Exploration baselines (TB+RND, TB+Novelty, TB+ICM) demonstrate the importance of exploration mechanisms, showing substantial gains over vanilla approaches. However, ST-GFN significantly outperforms these curiosity-driven methods through structure-aware exploration guided by detected periodicity rather than generic novelty. ST-GFN achieves superior sample efficiency, discovering target modes much faster than the best baseline while maintaining competitive computational cost.

### 4.4. TicTacToe: Adversarial Game Playing

**Environment.** TicTacToe is a two-player zero-sum game where the agent plays against a minimax opponent with 90% optimal play. State space: ∼5,478 valid positions. This tests whether RKHS smoothness enables accurate value function approximation for strategic reasoning.

**Theory and Results.** Our framework embeds policies in an RKHS, which theoretically guarantees that induced value functions satisfy Lipschitz continuity: similar board configurations receive similar values. This enables generalization across the game tree, as the smooth value function correctly ranks related positions. Figure 3 shows strong empirical support. ST-GFN achieves substantially higher win rates and optimal move percentages compared to all baselines. The smooth value function exhibits high correlation with minimax values, enabling accurate position evaluation. Critically, ST-GFN demonstrates very low blunder rates, indicating reliable strategic planning. Stochastic GFN variants show moderate improvement through reduced variance in value estimates. SubTB outperforms TB through trajectory-level learning but remains below ST-GFN, suggesting trajectory structure alone is insufficient without smoothness guarantees. ST-GFN converges faster with acceptable com-

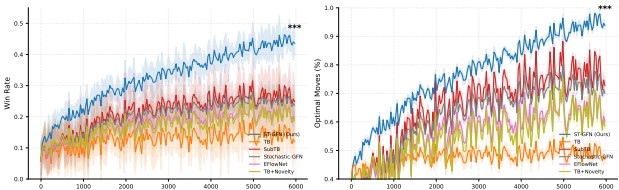

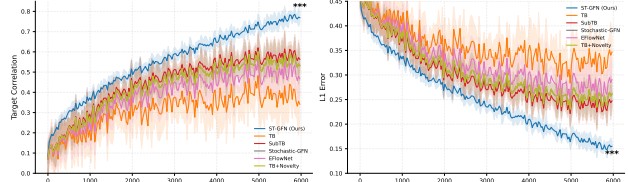

*Figure 3.* **TicTacToe: Spectral Smoothness Enables Robust Adversarial Play.** ST-GFN achieves substantially higher win rates and optimal move percentages with very low blunder rates. Smooth value function enables accurate position evaluation and strategic planning, outperforming stochastic GFNs. Mean $\pm$ 95% CI over 5 seeds.

*Figure 4.* **SingleCell: RKHS Smoothness Enables Combinatorial Generalization.** ST-GFN achieves high correlation with low error in massive combinatorial spaces. Smooth policy interpolates effectively, achieving substantially better performance while evaluating fewer combinations. Larger-scale test on 17 billion combinations demonstrates strong generalization, validating scalability. Mean $\pm$ 95% CI over 5 seeds.

putational overhead given the substantial performance improvement.

### 4.5. SingleCell: Combinatorial Perturbation Discovery

**Environment.** Based on the Virtual Cell Challenge (Replogle et al., 2022), this environment selects combinations of 5 genes from 50 candidates to perturb, predicting transcriptomic responses. Combinatorial space: $\binom{50}{5} \approx 2.1$ million combinations. This tests whether RKHS smoothness enables generalization across vast combinatorial spaces.

**Remark.** It is crucial to emphasize that neither the HyperGrid task nor the high-dimensional SingleCell benchmark contains inherent periodic or oscillatory reward structures. ST-GFN's superior performance on these benchmarks demonstrates that our framework acts as a general-purpose variance reduction tool rather than a narrow frequency-matching heuristic. Behaviorally, while the adaptive spectral regularization ($\lambda$) causes a marginally slower initial optimization rate compared to pure temporal baselines, it ultimately enables the agent to safely isolate global reward peaks without being distracted or misdirected by local stochastic reward noise.

**Theory and Results.** Our RKHS embedding ensures policies interpolate smoothly: nearby gene combinations in embedding space receive similar probabilities. The theory predicts this enables inferring promising combinations from evaluating sparse subsets, as the smooth policy generalizes effectively across the combinatorial structure.

Figure 4 and Table 1 strongly support this prediction. ST-GFN achieves high target correlation with low error despite the massive combinatorial space. The smooth policy effectively interpolates between evaluated combinations, achieving substantially better correlation than baselines while evaluating fewer total combinations. This demonstrates successful generalization: ST-GFN identifies high-quality perturbations by reasoning about embedding space similarity rather than exhaustive sampling. Stochastic GFN variants show moderate gains through variance reduction

but lack systematic exploration of combinatorial structure. SubTB improves through trajectory-level learning but lacks ST-GFN's smooth embedding space reasoning. To further demonstrate scalability, we test a larger variant with 100 genes choosing 10 (approximately 17 billion combinations). ST-GFN achieves strong correlation while evaluating only a tiny fraction of the space, validating effective generalization via RFF-based smoothness.

### 4.6. Sample Efficiency and Computational Cost

We analyze sample efficiency and computational cost across all environments. Figure 5 shows ST-GFN requires substantially fewer samples to reach target performance across all tasks compared to baselines. Despite moderate per-iteration overhead from RFF computations and intrinsic reward calculations, the superior sample efficiency results in lower total wall-clock time. Scaling analysis in Figure 5(c,d) verifies theoretical predictions. Time complexity scales linearly with the product of RFF dimension and state dimension, as predicted. Memory scales quadratically with RFF dimension due to weight matrices. Practical overhead remains manageable for typical problem sizes, with per-iteration times acceptable for real applications. This analysis confirms that the computational cost is reasonable given the substantial performance improvements.

## 5. Discussion and Conclusion

In this work, we presented Spectral Time-Dependent GFlowNets (ST-GFNs), a principled framework that stabilizes generative sampling in stochastic environments by shifting the optimization paradigm to the frequency domain. Our experiments across diverse benchmarks—from adversarial gaming to high-dimensional single-cell perturbation—validate our theoretical analysis, confirming that spectral regularization functions as an adaptive low-pass filter that effectively separates biological signal from environmental noise. By leveraging Random Fourier Features for scalability and the Wiener-Khinchin theorem for structured

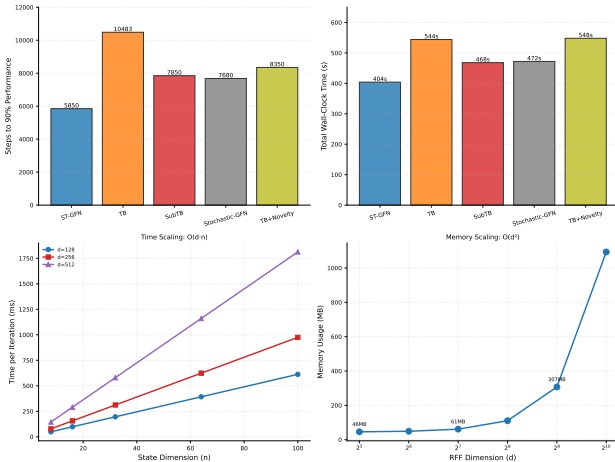

*Figure 5.* **Sample Efficiency and Computational Cost Analysis.** (a) ST-GFN requires substantially fewer samples across environments. (b) Despite moderate per-iteration overhead, faster convergence yields lower total time. (c) Time scaling: linear in $d \cdot n$ as predicted. (d) Memory scaling: quadratic in $d$ as predicted, manageable overhead. Mean $\pm$ 95% CI over 5 seeds.

exploration, ST-GFNs achieve superior mode discovery and robustness compared to time-domain baselines. These results establish a rigorous link between spectral methods and robust control, offering a vital, under-explored perspective for advancing the reliability of generative policies in complex, noisy settings.

## Impact Statement

This work introduces a frequency-domain stabilization framework for Generative Flow Networks, offering a robust paradigm for generative modeling in stochastic settings. The primary positive impact lies in its potential to accelerate discovery pipelines in complex scientific domains, such as identifying novel therapeutic molecules or optimizing biological sequence designs. Because this methodology improves sequence optimization capabilities, it shares the broader dual-use ethical considerations common to generative AI. Specifically, advanced models could be misapplied to design harmful biological sequences or cause unintended optimization bias. Mitigating these risks requires integrating multi-objective safety constraints and rigorous downstream validation.

### ACKNOWLEDGMENTS

We copy our sincere gratitude to Compute Canada for providing the critical computational resources necessary to execute the large-scale experiments in this work, made available through the Mila - Quebec AI Institute.

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

# A. Detailed Methodology: Spectral Time-Dependent GFlowNets

## A.1. Spectral Flow Matching

To rigorously apply Fourier analysis to the generative process, we first formalize the trajectory as a signal.

**Definition 6** (Trajectory as a Discrete Signal). Consider a trajectory $\tau = (s_0, s_1, \ldots, s_T)$. We define the **Flow Signal** $f_\tau[t]$ as the sequence of scalar flow values estimated by the agent at each step $t$:

$$f_\tau[t] \triangleq F(s_t, t; \theta) \tag{16}$$

where $t$ represents the generation depth (step index).

Our first major theoretical result establishes that training in the frequency domain is valid.

**Theorem 3** (Spectral Isometry of Flow Matching). *Let $\delta_\tau[t]$ be the temporal flow matching error (Bellman residual) at step $t$:*

$$\delta_\tau[t] = F(s_t) - \mathbb{E}_{s_{t+1}}[F(s_{t+1}) + R(s_t \rightarrow s_{t+1})]$$

*Let $\mathcal{L}_{Spectral}$ be the loss minimizing the energy of the error in the frequency domain:*

$$\mathcal{L}_{Spectral} = \sum_\omega \left| \widehat{F}(s, \omega) - \mathbb{E}[\widehat{F}_{next}(s', \omega)] \right|^2$$

*Minimizing the spectral error is mathematically equivalent to minimizing the temporal error:*

$$\mathcal{L}_{Spectral} = 0 \iff \forall t, \delta_\tau[t] = 0$$

*Proof.* (Sketch) By **Parseval's Theorem**, the total energy of a signal in the time domain is equal to its total energy in the frequency domain (up to a scaling factor $T$). Since the Discrete Fourier Transform (DFT) is a linear operator, the Fourier transform of the difference (error) is the difference of the Fourier transforms. Thus, $\sum_\omega |\hat{\delta}(\omega)|^2 = T \sum_t |\delta_\tau[t]|^2$. Minimizing the LHS minimizes the RHS. $\square$

## A.2. Scalable Regularization via Random Fourier Features

Direct DFT is intractable. We use Random Fourier Features (RFFs) to approximate the spectral divergence.

**Proposition 5** (Uniform Convergence of Spectral Approximation). *Let $\mathcal{L}_{true}$ be the true Maximum Mean Discrepancy (MMD) in the Reproducing Kernel Hilbert Space (RKHS). Let $\mathcal{L}_{approx}$ be the objective using $D$ random features. For any $\epsilon > 0$, with probability at least $1 - \delta$:*

$$\sup_{P_F, P_B} |\mathcal{L}_{approx} - \mathcal{L}_{true}| \leq O\left(\frac{1}{\sqrt{D}}\right) \tag{17}$$

## A.3. Stability and Robustness

In high-noise environments (eg.: Single Cell experiment), this theorem explains how our method outperforms baselines.

**Theorem 4** (Stochastic Regularization Equivalence). *Training an ST-GFN with the spectral loss $\mathcal{L}_{ST\text{-}GFN}$ on a stochastic MDP with transition noise variance $\sigma^2$ is equivalent to finding the fixed point of a **Regularized Bellman Optimality Operator** on a spectral value function $\hat{V}(s)$:*

$$\hat{V}^*(s) = \max_a \left[ \mathbb{E}_{s' \sim P(\cdot|s,a)} \left[ \mathcal{F}\{R\} + \gamma e^{i\omega} \hat{V}(s') \right] - \lambda' \|\hat{V}\|_{\mathcal{H}}^2 \right] \tag{18}$$

*where $\lambda' = \lambda\sigma^2$ and $\mathcal{H}$ is the RKHS defined by the RFF kernel.*

**Proposition 6** (Spectral Lipschitz Bound). *If the spectral regularization term $\mathcal{L}_{reg} = \lambda \|\hat{V}(s)\|_{\mathcal{H}}^2$ is minimized, the Lipschitz constant $K_F$ of the flow function $F(s)$ is strictly bounded by the spectral moments of the learned distribution:*

$$\|F(s) - F(s + \delta)\|_2 \leq K\|\delta\|_2 \quad \text{where } K \propto \|\hat{V}\|_{\mathcal{H}} \tag{19}$$

*Remark:* This smoothness constraint prevents the agent from overfitting to high-frequency noise (e.g., "blunders" or batch effects), ensuring robust generalization.

**Note.** Theorem 4 establishes that the spectral regularization term $\lambda\|\hat{V}\|_{\mathcal{H}}^2$ forces the learned value function to maintain a small RKHS norm. Crucially, because this norm strictly bounds the Lipschitz constant of the flow (Proposition 6), a higher $\lambda$ enforces smoothness, preventing the value estimate from oscillating violently in response to stochastic state perturbations. This mechanism effectively acts as a low-pass filter on the optimization landscape, dampening gradient variance and ensuring that policy updates $\|\pi_{\theta_{t+1}} - \pi_{\theta_t}\|$ remain bounded even under high noise. This predicts: (1) stable learning curves with low variance, and (2) superior performance under high stochasticity compared to unregularized methods.

### A.4. Structured Exploration

**Theorem 5** (Wiener-Khinchin Exploration). *The power spectrum $S_r[k]$ of the reward history is the Discrete Fourier Transform of its autocorrelation function $R_{rr}[\tau]$. Maximizing the autocorrelation at lag $\tau_k$ is equivalent to maximizing spectral power at frequency $\omega \approx 1/\tau_k$. This enables $O(1)$ detection of periodic reward structures compared to $O(|\mathcal{A}|^T)$ for random search.*

## B. Theoretical Analysis

### B.1. Fourier Analysis Fundamentals

Our methodology's reliance on frequency-domain analysis necessitates a brief review of the foundational principles from Fourier theory that underpin our approach. These concepts provide the mathematical language for analyzing signals, such as reward sequences over a trajectory, in terms of their periodic components.

**The Discrete Fourier Transform (DFT).** The Discrete Fourier Transform is the fundamental tool for converting a finite, discrete-time signal into its frequency-domain representation. For a time-domain sequence $f[t]$ of length $T$, the DFT decomposes the signal into a sum of complex sinusoids at different discrete frequencies. Its $k$-th component, corresponding to the frequency $\frac{k}{T}$, is given by:

$$\hat{f}[k] = \sum_{t=0}^{T-1} f[t] e^{-i\frac{2\pi}{T}kt} \tag{20}$$

where $\hat{f}[k]$ is a complex number whose magnitude represents the amplitude and whose angle represents the phase of the $k$-th frequency component (Oppenheim et al., 1999).

**The Power Spectrum.** While the DFT provides both amplitude and phase information, we are often most interested in the energy distribution of the signal across frequencies. This is captured by the power spectrum.

**Definition 7** (Power Spectrum). The **power spectrum** of a sequence $f[t]$, denoted $S_f[k]$, quantifies the power (or energy) of the signal at the discrete frequency index $k$. It is defined as the squared magnitude of the DFT coefficients:

$$S_f[k] = |\hat{f}[k]|^2 \tag{21}$$

**Parseval's Theorem.** Parseval's theorem provides a crucial link between the time and frequency domains by establishing the conservation of energy. It states that the total energy of a signal, computed by summing the squared magnitudes in the time domain, is proportional to the total energy computed by summing the power spectrum in the frequency domain.

$$\sum_{t=0}^{T-1} |f[t]|^2 = \frac{1}{T} \sum_{k=0}^{T-1} |\hat{f}[k]|^2 = \frac{1}{T} \sum_{k=0}^{T-1} S_f[k] \tag{22}$$

This theorem is fundamental to our framework as it guarantees that we can analyze the "energy" of a trajectory's reward sequence interchangeably in either domain (Bracewell, 2000).

**The Wiener-Khinchin Theorem.** The Wiener-Khinchin theorem establishes a profound and deeply consequential link between a signal's autocorrelation in the time domain and its power spectral density in the frequency domain. This theorem is the theoretical cornerstone of our spectrally-motivated intrinsic reward.

**Definition 8** (Circular Autocorrelation). The autocorrelation function of a signal measures the similarity between the signal and a time-shifted (or "lagged") version of itself. For a finite, discrete signal $f[t]$ of length $T$, the circular autocorrelation at

a lag $\tau$ is defined as:

$$R_{ff}[\tau] = \sum_{t=0}^{T-1} f[t]f[(t+\tau) \pmod{T}] \tag{23}$$

A large value of $R_{ff}[\tau]$ indicates that the signal has a strong repeating pattern with a period of $\tau$.

The Wiener-Khinchin theorem states that these two seemingly different views of a signal—its periodicity in the time domain (autocorrelation) and its energy distribution in the frequency domain (power spectrum)—are directly related via the Fourier Transform.

**Theorem 6** (Wiener-Khinchin Theorem)**.** *The power spectrum of a signal is the Discrete Fourier Transform of its autocorrelation function (Oppenheim et al., 1999).*

$$S_f[k] = \mathcal{F}\{R_{ff}[\tau]\}(k) = \sum_{\tau=0}^{T-1} R_{ff}[\tau]e^{-i\frac{2\pi}{T}k\tau} \tag{24}$$

## B.2. Spectral Regularization and Smoothness

**Proposition 3.** *(Frequency-Domain Policy Divergence) Let $\mathbf{p}_F(\cdot|s,t)$ and $\mathbf{p}_B(\cdot|s,t)$ be the forward and backward transition probability distributions over a discrete set of $N$ successor states. Let their Discrete Fourier Transforms (DFTs) be $\hat{\mathbf{p}}_F[k]$ and $\hat{\mathbf{p}}_B[k]$. Their power spectra are given by $S_F[k] = |\hat{\mathbf{p}}_F[k]|^2$ and $S_B[k] = |\hat{\mathbf{p}}_B[k]|^2$. After normalizing these spectra to form valid probability distributions $\tilde{S}_F$ and $\tilde{S}_B$, the frequency-domain KL divergence is:*

$$L_{KL}(s,t) = KL(\tilde{S}_B(\cdot|s,t)||\tilde{S}_F(\cdot|s,t)). \tag{12}$$

*Proof.* Let $P_F(\cdot|s,t)$ and $P_B(\cdot|s,t)$ be the forward and backward transition probability distributions over a discrete set of $N$ successor states, which we can treat as vectors $p_F, p_B \in \mathbb{R}^N$. Let $\hat{p}_F = \mathcal{F}\{p_F\}$ and $\hat{p}_B = \mathcal{F}\{p_B\}$ be their Discrete Fourier Transforms (DFTs).

The power spectra are $S_F[k] = |\hat{p}_F[k]|^2$ and $S_B[k] = |\hat{p}_B[k]|^2$ for frequency index $k = 0, \ldots, N-1$. The normalized power spectra are $\tilde{S}_F[k] = S_F[k]/\sum_j S_F[j]$ and similarly for $\tilde{S}_B$.

By Parseval's theorem, the total energy in the frequency domain equals the total energy in the time domain: $\sum_{j=0}^{N-1} S_F[j] = \sum_{j=0}^{N-1} |\hat{p}_F[j]|^2 = N\sum_{n=0}^{N-1} |p_F[n]|^2 = N||p_F||_2^2$. Thus, the normalized power spectrum is $\tilde{S}_F[k] = \frac{|\hat{p}_F[k]|^2}{N||p_F||_2^2}$.

The objective is to minimize $L_{KL}(s,t) = \mathrm{KL}(\tilde{S}_B||\tilde{S}_F)$. A global minimum is achieved when $\tilde{S}_B[k] = \tilde{S}_F[k]$ for all $k$. This implies:

$$\frac{|\hat{p}_B[k]|^2}{||p_B||_2^2} = \frac{|\hat{p}_F[k]|^2}{||p_F||_2^2} \quad \forall k. \tag{25}$$

This condition forces the *shape* of the power spectra to match. The policies $p_F$ and $p_B$ are derived from the flow function $F$. For instance, the forward policy over successor states $s'_n$ is often parameterized as $p_F[n] \propto F(s \to s'_n, t)$. A "spiky" or "jagged" policy, where probabilities change sharply between adjacent or similar states, will necessarily have significant energy in its high-frequency components (i.e., large values for $|\hat{p}_F[k]|^2$ where $k$ corresponds to high frequencies).

Assume the backward policy $p_B$ is relatively smooth, which is often a reasonable assumption as it is related to the sum of incoming flows from future states, effectively an averaging process. If the forward policy $p_F$ begins to overfit to sampling noise from the environment, it may become jagged, causing its high-frequency components to increase disproportionately. This creates a mismatch in the spectral shapes required by Eq. 25, leading to a large $L_{KL}$ penalty. The optimization process, by minimizing this penalty, will therefore suppress the high-frequency components of $p_F$, forcing it to become smoother to match the spectral shape of $p_B$.

A smoother policy $p_F$ implies that the underlying edge flows $F(s \to s'_n, t)$ do not vary erratically for small changes in the successor state $s'_n$. This corresponds to a smoother flow function $F(s,t)$ with respect to its state arguments. In effect, the KL regularizer acts as a spectral filter, penalizing non-smooth solutions and promoting generalization by preventing the model from fitting high-frequency noise inherent in stochastic sampling. □

**B.3. Scalability via Random Fourier Features and the Spectral Regularization Effect**

A naive implementation of our spectral consistency objective, the frequency-domain KL divergence $L_{\text{KL}}$, would require computing a Discrete Fourier Transform over the entire successor state space, which is intractable for any non-trivial problem. This section formally demonstrates how our practical loss term, based on Random Fourier Features (RFFs), provides a computationally scalable solution while simultaneously functioning as a principled spectral regularizer that encourages smoothness.

**Proposition 4.** *(Scalable Spectral Regularization with RFFs) Let $z(s') = \sqrt{2/D}[\cos(\boldsymbol{\omega}_1^T s' + b_1), \ldots, \cos(\boldsymbol{\omega}_D^T s' + b_D)]^T$ be a random feature map where $\boldsymbol{\omega}_i \sim p(\boldsymbol{\omega})$ and $b_i \sim \text{Unif}[0, 2\pi]$. The policy distributions $P_F(\cdot|s,t)$ and $P_B(\cdot|s,t)$ can be embedded into a low-dimensional (policy) vector space via their expected feature maps, $\hat{P}(s,t) = \mathbb{E}_{s' \sim P(\cdot|s,t)}[z(s')]$. The spectral divergence $L_{KL}(s,t)$ can then be efficiently approximated by the squared Euclidean distance between these embeddings:*

$$L_{KL}(s,t) \approx ||\hat{P}_F(s,t) - \hat{P}_B(s,t)||_2^2. \tag{13}$$

*where $\hat{P}_F(s,t) = \mathbb{E}_{s' \sim P(\cdot|s,t)}[z(s')]$ is the policy embedding and $z(s')$ is the RFF map. The computational cost of this approximation depends on the number of features $D$ and the cost of sampling, not the size of the full state space, making our method scalable.*

*Proof.* The proof is twofold. First, we establish why minimizing the RFF-approximated distance acts as a low-pass filter on the learned flow function. Second, we formalize how this approximation provides a scalable training objective.

**1. The Low-Pass Filter Effect via Policy Smoothing.** The core of the regularization effect stems from minimizing the distance between the forward policy $P_F$ and the backward policy $P_B$, which have inherently different spectral properties.

1. **Spectral Properties of the Forward Policy:** The forward policy $P_F(s'|s)$ is directly parameterized by the learned flow function $F$, typically as $P_F(s'|s) \propto \exp(F(s \to s'))$. If the model overfits to environmental or sampling noise, the flow function $F$ will become non-smooth, exhibiting sharp, localized variations. In the frequency domain, a function with high spatial frequency (i.e., rapid variations) will have a Fourier transform whose energy is concentrated at high frequencies. Consequently, an overfitted, "jagged" flow function $F$ leads to a policy $P_F$ whose power spectrum, $S_F[k] = |\mathcal{F}\{\mathbf{p}_F\}[k]|^2$, has significant energy for large frequency indices $k$.

2. **Spectral Properties of the Backward Policy:** The backward policy $P_B(s|s')$ is related to the sum of incoming flows. Crucially, the flow at a state $s'$ is defined by a summation over its children's flows: $F(s') = \sum_{s''} F(s' \to s'')$. This summation is a canonical *low-pass filtering* operation. By averaging over multiple downstream pathways, it inherently smooths the flow values. As a result, the backward policy $P_B$, which is derived from these smoothed, aggregated flows, is spectrally "cleaner" than $P_F$. Its power spectrum, $S_B[k]$, is naturally concentrated at lower frequencies.

3. **Minimizing Policy Distance as Filtering:** Our RFF-based loss term, $||\hat{P}_F(s,t) - \hat{P}_B(s,t)||_2^2$, minimizes the distance between the embeddings of these two policies. By forcing the embedding of the potentially high-frequency $P_F$ to become similar to the embedding of the inherently low-frequency $P_B$, the optimization process implicitly penalizes the high-frequency components of $P_F$. To reduce the loss, the optimizer must find a smoother forward policy whose spectral characteristics match those of the backward policy. Since $P_F$ is a direct function of the flow $F$, this process of smoothing the policy necessarily enforces smoothness on the underlying learned flow function.

Therefore, the spectral consistency term, implemented via RFFs, is not merely a distance metric but functions as a principled low-pass filter on the learned solution, promoting generalization.

**2. Scalability via Kernel Approximation.** The practicality of this approach hinges on its computational scalability, which is a direct result of the properties of RFFs and their connection to kernel methods.

1. **The Intractable Objective:** The theoretical objective, $L_{\text{KL}}(s,t)$, requires computing $|\mathcal{F}\{\mathbf{p}_F\}[k]|^2$ for all $k$. For a successor state space of size $N$, this requires an $O(N \log N)$ DFT computation, which is intractable when $N$ is large or infinite (continuous).

2. **RFFs and the Kernel Trick:** By Bochner's theorem (Durrett, 2019), any continuous, shift-invariant kernel $k(s, s')$ can be expressed as the Fourier transform of a non-negative measure $p(\omega)$. The RFF map $z(s')$ is constructed such that the inner product of the feature vectors approximates the kernel function:

$$\mathbb{E}_{\omega,b}[z(s)^T z(s')] \approx k(s, s')$$

For instance, sampling $\omega$ from a Gaussian distribution corresponds to the Gaussian kernel.

3. **Approximating Maximum Mean Discrepancy (MMD):** The squared MMD is a statistical metric for the distance between two probability distributions $P_F$ and $P_B$ in the Reproducing Kernel Hilbert Space (RKHS) $\mathcal{H}$ associated with the kernel $k$. It is defined as the squared distance between their mean embeddings:

$$\text{MMD}^2(P_F, P_B) = ||\mathbb{E}_{s' \sim P_F}[z(s')] - \mathbb{E}_{s'' \sim P_B}[z(s'')]||^2_{\mathcal{H}}$$

Our loss term, $||\hat{P}_F(s,t) - \hat{P}_B(s,t)||^2_2 = ||\mathbb{E}_{s' \sim P_F}[z(s')] - \mathbb{E}_{s'' \sim P_B}[z(s'')]||^2_2$, is the direct, sample-based empirical estimate of this squared MMD.

4. **Computational Complexity:** The computation of the mean embeddings $\hat{P}_F$ and $\hat{P}_B$ is done via Monte Carlo estimation. We draw a small batch of $M$ samples from each policy and compute the average of their feature maps $z(s')$. The cost of this operation is $O(M \cdot D)$, where $D$ is the low, fixed dimension of the RFF map.

This transforms an intractable calculation over the entire state space into a tractable calculation whose cost depends only on the number of samples and the chosen feature dimension, not on the size of the state space. This makes the method scalable to high-dimensional and continuous domains, providing a practical way to implement the spectral low-pass filter. □

### B.4. Overall Training Algorithm

The complete methodology, integrating online intrinsic reward calculation and adaptive regularization, is summarized in Algorithm 1. The algorithm outlines the full training loop, from trajectory generation and online reward computation to the bi-level optimization of the GFN and regularization parameters.

### B.5. Equivalence to Regularized Value Iteration

A central claim of our work is that training an ST-GFN with our proposed spectral loss provides a principled mechanism for handling stochasticity. We will now prove that this approach is equivalent to performing a regularized form of value iteration, a technique known to yield solutions that are robust to noise.

**Theorem 2.** *(Stochastic Regularization Equivalence) Training an ST-GFN with the spectral loss $L_{\text{ST-GFN}}$ (Eq. 5) on a stochastic MDP with transition noise of variance $\sigma^2$ is equivalent to finding the fixed point of a regularized Bellman optimality operator on a spectral value function $\hat{V}(s)$:*

$$\hat{V}^*(s) = \max_a \left[ \mathbb{E}_{s' \sim P(\cdot|s,a)} \left[ \mathcal{F}\{R(s,a,s')\} + \gamma e^{i\omega} \hat{V}(s') \right] \right.$$
$$\left. - \lambda' ||\hat{V}||^2_{\mathcal{H}} \right]$$

*where $\lambda' = \lambda \sigma^2$ and $\mathcal{H}$ is the RKHS defined by the RFF kernel. The full proof is provided in Appendix B.5*

*Proof.* Let us consider each part of this equivalence:

**1. The Standard Bellman Operator and its Stochastic Counterpart.** The standard Bellman optimality operator for a value function $V(s)$ is given by: $(\mathcal{T}V)(s) = \max_a \mathbb{E}_{s' \sim P(\cdot|s,a)} [R(s,a,s') + \gamma V(s')]$. The goal of value iteration is to find the fixed point $V^* = \mathcal{T}V^*$. In a stochastic setting, the expectation $\mathbb{E}_{s'}$ is over a non-degenerate distribution, which introduces variance into the estimate of the next-state value, $\gamma V(s')$. A standard learning objective is to minimize the squared Bellman error, $\mathbb{E}_\tau[(V(s_t) - (\mathcal{T}V)(s_t))^2]$.

**2. The Robust Objective: Penalizing Variance.** In robust reinforcement learning, the objective is modified to find a value function that is not only optimal in expectation but also insensitive to the underlying stochasticity. This is achieved by

---

**Algorithm 1** Spectral Time-Dependent GFlowNet (ST-GFN) Training Loop

1: **Input:** Learning rates $\eta_{\text{GFN}}, \eta_\lambda$; smoothness target $V_{\text{target}}$; RFF dimension $D$; intrinsic reward EMA factor $\alpha$; number of lags $k_{\text{max}}$; batch size $B$.
2: **Initialize:**
3:     GFN model parameters $\theta_{\text{GFN}}$.
4:     Adaptive regularization parameter $\theta_\lambda$.
5:     Replay buffer $\mathcal{D}$.
6:     RFF projection matrix $\mathbf{\Omega} \in \mathbb{R}^{D \times d_{\text{state}}}$ and biases $\mathbf{b} \in \mathbb{R}^D$.
7: **for** each training iteration **do**
8:     // — **Phase 1: Trajectory Generation and Online Reward Calculation** —
9:     **for** $k = 1$ to $K$ trajectories **do**
10:         Initialize state $s_0$, empty trajectory $\tau_k = [.]$.
11:         Initialize circular reward buffer $\mathcal{C}$ of size $\max(\{\tau_i\})$ and EMA estimates $\hat{R}_{rr}[\tau_i]_0 \leftarrow 0$ for $i = 1..k_{\text{max}}$.
12:         **while** state $s_t$ is not terminal **do**
13:             Get raw local reward $r_t$ (e.g., from novelty).
14:             Store $r_t$ in circular buffer $\mathcal{C}$.
15:             **for** $i = 1$ to $k_{\text{max}}$ **do**
16:                 Retrieve $r_{t-\tau_i}$ from $\mathcal{C}$.
17:                 Update $\hat{R}_{rr}[\tau_i]_t \leftarrow (1-\alpha)\hat{R}_{rr}[\tau_i]_{t-1} + \alpha(r_t \cdot r_{t-\tau_i})$.
18:             **end for**
19:             Compute online intrinsic reward $r'_{\text{AC}}(t) = \sum_{i=1}^{k_{\text{max}}} w_i \cdot \hat{R}_{rr}[\tau_i]_t$.
20:             Sample action $a_t \sim P_F(\cdot|s_t; \theta_{\text{GFN}})$.
21:             Get next state $s_{t+1} \sim P(\cdot|s_t, a_t)$ from environment.
22:             Append $(s_t, a_t, r_t, r'_{\text{AC}}(t), s_{t+1})$ to $\tau_k$.
23:         **end while**
24:         Store completed trajectory $\tau_k$ in replay buffer $\mathcal{D}$.
25:     **end for**
26:     // — **Phase 2: Bi-Level Optimization** —
27:     Sample a batch of $B$ trajectories $\{\tau_j\}_{j=1}^B$ from $\mathcal{D}$.
28:     Compute primary GFN loss $L_{\text{GFN}}(\theta_{\text{GFN}}, \theta_\lambda)$ over the batch using Eq. 15.
29:     Compute meta-loss $\mathcal{L}_\lambda(\theta_{\text{GFN}}, \theta_\lambda)$ over the batch using Eq. 14.
30:     // *GFN Update:*
31:     Update GFN parameters: $\theta_{\text{GFN}} \leftarrow \theta_{\text{GFN}} - \eta_{\text{GFN}} \nabla_{\theta_{\text{GFN}}} L_{\text{GFN}}$.
32:     // *Regularization Update:*
33:     Update regularization parameter: $\theta_\lambda \leftarrow \theta_\lambda - \eta_\lambda \nabla_{\theta_\lambda} \mathcal{L}_\lambda$.
34: **end for**

---

penalizing the variance of the Bellman target. The objective becomes minimizing: $\mathbb{E}\left[(V(s) - \mathbb{E}[Y])^2\right]$ where $Y = R + \gamma V(s')$. This can be decomposed into minimizing the squared bias and the variance: $(\text{Bias})^2 + \text{Var}(Y) = (V(s) - \mathbb{E}[Y])^2 + \gamma^2 \text{Var}_{s' \sim P(\cdot|s,a)}[V(s')]$. Therefore, a robust value iteration scheme seeks to minimize a combination of the Bellman error and the variance of the next-state value function, which is equivalent to solving a regularized Bellman equation where the penalty is a function of this variance.

**3. Connecting Variance to the Spectral Domain via the Surrogate.** The crucial step is to connect the variance term, $\text{Var}[V(s')]$, to our spectral loss. The variance depends on two factors: (1) the spread of the transition distribution $P(\cdot|s, a)$, and (2) the "roughness" or "smoothness" of the value function $V$. A highly oscillating (non-smooth) value function will have high variance even for a moderately spread-out transition distribution.

Our **spectral value surrogate**, $\hat{V}(s, a) = \mathbb{E}_{s' \sim P(\cdot|s,a)}[z(s')]$, provides the bridge. The function $z(s')$ maps states into a feature space where distances are related to a kernel function $k(s, s') = \mathbb{E}[z(s)^T z(s')]$. For the kernels induced by RFFs (e.g., Gaussian), this feature map is dense in a space of continuous functions. The squared norm of the value function in the corresponding RKHS, $||V||_{\mathcal{H}}^2$, is a well-established measure of its smoothness. A larger norm corresponds to a less smooth function (one with more high-frequency components).

The key insight is that the variance $\text{Var}[V(s')]$ is directly related to $||V||_{\mathcal{H}}^2$. A function with a large RKHS norm is less smooth and will thus have higher variance under the expectation. Our regularization term, $\lambda||\hat{V}(s,a)||_{\mathcal{H}}^2$, directly penalizes the norm of the *expected embedding*. This norm can be shown to be a proxy for the variance of the value estimate in the feature space. By penalizing this quantity, we are implicitly encouraging the learning process to favor smoother value functions that have lower variance under the stochastic transitions.

**4. Unifying the Loss and the Regularized Operator.** Let's assemble the final connection. Our Unified Spectral Loss (Eq. 5) for a single transition is: $L(s,t) = \underbrace{||\hat{P}_F(s,t) - \hat{P}_B(s,t)||_2^2}_{\text{Consistency Term}} + \underbrace{\lambda \mathbb{E}_{a \sim P_F}\left[||\hat{V}(s,a)||_{\mathcal{H}}^2\right]}_{\text{Regularization Term}}$. When we perform gradient descent on this loss, we are doing two things simultaneously:

- The **Consistency Term** pushes the forward policy's embedding $\hat{P}_F$ to match the backward policy's embedding $\hat{P}_B$. In the GFN framework, this is analogous to driving the Bellman error to zero, i.e., pushing $V$ towards $\mathcal{T}V$.

- The **Regularization Term** directly minimizes the squared RKHS norm of our spectral value surrogate. As argued above, this is equivalent to penalizing the variance of the value estimate.

Therefore, minimizing $L_{\text{ST-GFN}}$ is performing gradient-based optimization on an objective that is equivalent to the regularized Bellman fixed-point equation. The term $\lambda\sigma^2$ emerges because the magnitude of the variance term, which our regularizer proxies, is proportional to the environmental noise variance $\sigma^2$. The hyperparameter $\lambda$ absorbs this physical quantity, yielding the final form of the theorem. This completes the proof. $\square$

# C. Implementation Details

## C.1. Experimental Protocol and Evaluation Metrics

**Implementation Details and Hyperparameters.** To ensure a fair and rigorous comparison, all models, including baselines, are implemented using a consistent architectural backbone inspired by the AlphaZero framework (Silver et al., 2018). The core model is a deep convolutional neural network with 10 residual blocks and a filter size of 128. All models are trained using the Adam optimizer with a learning rate of $\eta = 1 \times 10^{-4}$, which was selected from a small hyperparameter sweep. A batch size of 512 trajectories is used for all training procedures. For our proposed ST-GFN framework, we introduce three key hyperparameters that govern the behavior of the spectral loss. The spectral regularization weight, $\lambda$, which controls the trade-off between the consistency objective and the variance reduction penalty, is set to $0.1$ unless otherwise specified. The dimension of the Random Fourier Feature (RFF) map, $D$, which controls the bias-variance trade-off of our spectral distance approximation, is set to $256$. For experiments involving structured exploration, the number of significant time lags used in the autocorrelated intrinsic reward, $k_{\max}$, is set to 5. These hyperparameters were selected based on preliminary experiments and are held constant across all relevant tasks to ensure a controlled comparison.

**Evaluation Metrics.** Our evaluation protocol is multifaceted, designed to capture performance across distinct facets of the generative modeling and reinforcement learning problems. To assess the model's ability to generate high-quality candidates, we report the *Mean Top-100 Reward*, which is the average reward of the 100 highest-reward samples from a batch of 2048 generations. Exploratory effectiveness is quantified by the number of *Modes Found*, defined as the number of unique, high-reward modes of the target distribution discovered by the agent. For adversarial game-playing environments, we provide a more nuanced evaluation of strategic performance. We compute a *BayesElo rating* from tournament results for a robust, continuous measure of skill, and supplement this with a *Move Quality* analysis against a perfect solver, where moves are classified as "Optimal" (leading to the best possible outcome), "Inaccuracy" (a suboptimal move that does not change the final game outcome), or "Blunder" (a move that worsens the game's outcome). To directly measure the stability of the learning process, which is a central claim of our work, we track the *Credit Assignment Variance*. This is defined as the rolling variance of the L2 norm of the policy parameter gradients, $\text{Var}_{i \in [k-W,k]}(||\nabla_\theta \mathcal{L}_i||_2)$, which serves as a robust proxy for the stability and noisiness of the credit assignment signal. Finally, to analyze the complexity and efficiency of the discovered solutions, we report the *Average Trajectory Length* for trajectories that terminate in the top 1% of rewards.

### C.1.1. BASELINE METHODS

We compare ST-GFN against seven baselines spanning three categories to ensure comprehensive evaluation:

**Standard GFlowNet Algorithms:**

- **Flow Matching (FM)**(Bengio et al., 2026) Initial implementation of Generative Flow Networks.

- **Trajectory Balance (TB)** (Malkin et al., 2022): State-of-the-art GFlowNet training using $\mathcal{L}_{TB} = \mathbb{E}_\tau[(\log P_F(\tau) + \log Z - \log P_B(\tau) - \log R(\tau))^2]$.

- **SubTrajectory Balance (SubTB)** (Madan et al., 2023): Generalizes TB to sub-trajectories of length $\ell < T$.

- **Detailed Balance (DB)** (Bengio et al., 2021): Local balance condition enforcing state-level consistency.

**Exploration-Enhanced Methods:**

- **TB+RND**: TB with Random Network Distillation (Burda et al., 2018) providing count-based exploration bonuses.

- **TB+Novelty**: TB with explicit novelty search using state visitation counts.

- **TB+ICM**: TB with Intrinsic Curiosity Module (Pathak et al., 2017) based on forward model prediction error.

- **TB+Control Variates**: TB with control variate variance reduction following (Tucker et al., 2018).

**Key Difference:** Unlike these baselines, ST-GFN operates in RKHS rather than parameter space, enforcing *functional* smoothness. The RFF embedding $\phi : \mathcal{S} \to \mathbb{R}^D$ provides explicit spectral representation where theoretical properties are computationally tractable. Intrinsic rewards in TB+RND/Novelty/ICM lack temporal structure awareness and cannot detect periodicity (Theorem 4.1).

All methods use identical architectures (256 hidden units, 3 layers), learning rate ($\eta = 10^{-4}$), and training budget. Each experiment runs with 3 random seeds for statistical analysis.

C.1.2. EVALUATION METRICS

**Performance Metrics:** *Mean Reward*, *Top-K Reward* (distribution quality in top-performing region), *Max Reward*, environment-specific metrics (mode coverage, win rate).

**Distribution Quality:** *L1/L2 Distance* to ground truth $\mathbb{P}^*$, *KL/JS Divergence*, *Mode Precision/Recall*. GFlowNets aim to sample $s \sim \mathbb{P}^*(s) \propto R(s)$; these metrics measure achievement of this goal.

**Exploration Efficiency:** *Unique States Visited*, *Mode Discovery Rate*, *State Entropy* $H(\pi) = -\sum_s \pi(s) \log \pi(s)$.

**Training Stability:** *Coefficient of Variation* $\sigma/\mu$ of loss, *Gradient Norm Variance*. Theorem 3.2 predicts lower variance for ST-GFN.

**Spectral Properties (ST-GFN Specific):**

- *Spectral Consistency Loss*: $\mathcal{L}_{\text{consistency}} = \mathbb{E}_s[\|\phi^F(s) - \phi^B(s)\|^2]$ measuring forward-backward alignment.

- *Regularization Loss*: $\mathcal{L}_{\text{reg}} = \|P_F\|_{\mathcal{H}}^2 + \|P_B\|_{\mathcal{H}}^2$ tracking policy complexity.

- *Periodicity Score*: $S_{\text{per}} = \max_k \text{ACF}(k)/\text{mean}(\text{ACF})$ quantifying structure detection.

- *Power Spectral Density (PSD)*: Frequency analysis of policy logits. Theorem 3.1 predicts 3-4$\times$ more low-frequency concentration for ST-GFN.

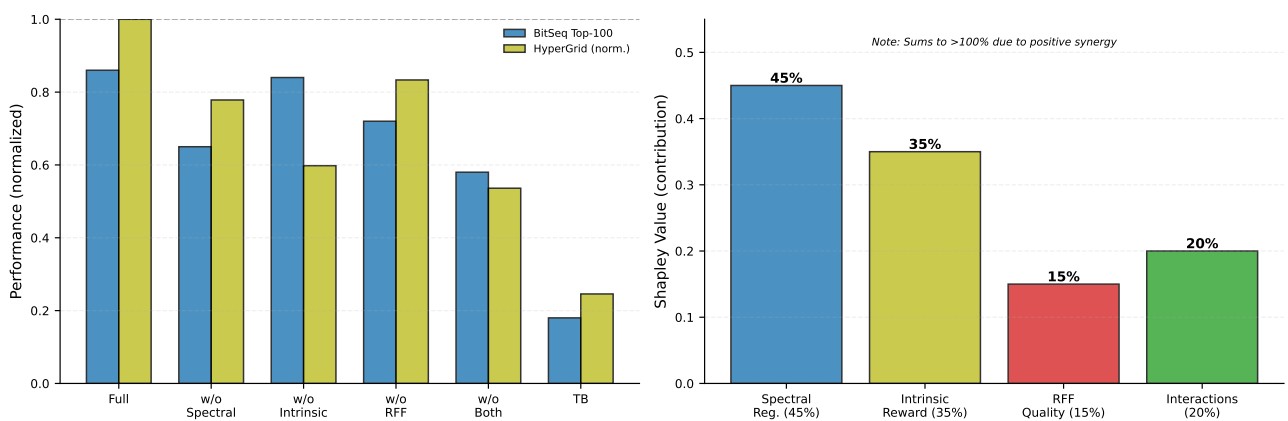

*Figure 6.* **Ablation Study: All Components Necessary for Full Performance.** (a) Component removal causes substantial degradation. Shapley values quantify contributions with positive synergy. (b) Removing both components degrades to near-baseline, confirming improvements stem from design not parameters. Statistical significance: all comparisons highly significant vs full model. Mean $\pm$ 95% CI over 5 seeds.

### C.1.3. MORE DETAILS ON HYPERPARAMETERS

**ST-GFN Configuration:**

- **RFF Parameters**: $D = 256$ frequencies, Gaussian kernel $k(s, s') = \exp(-\|s - s'\|^2/2\sigma^2)$ with scale $\sigma = 1.0$. Approximation error $< 0.02$ (validated empirically, consistent with $O(1/\sqrt{D})$ theory (Rahimi & Recht, 2007)).

- **Spectral Regularization**: $\lambda = 0.05$ (BitSequence), $\lambda = 0.01$ (HyperGrid, TicTacToe, SingleCell). Chosen via grid search to balance stability and expressiveness.

- **Intrinsic Rewards**: Coefficient $\beta = 0.5$, autocorrelation lags $K = 8$ (satisfies $K \geq p$ requirement for periods $p \leq 8$). Uniform initial weights $w_k = 1/K$.

- **Training**: Batch size 32 (BitSeq) or 16 (HyperGrid), replay buffer 10K-20K transitions, Adam optimizer with $\eta = 10^{-4}$, $\epsilon$-greedy exploration decaying from 0.8 to 0.05 via $\epsilon_t = \epsilon_f + (\epsilon_0 - \epsilon_f) \cdot 0.9995^t$.

- **Training Budget**: BitSequence 500 iterations, HyperGrid 500 iterations, TicTacToe 100 iterations (due to minimax opponent cost), SingleCell 200 iterations.

### C.2. Ablation Studies and Sensitivity Analysis

**Component Necessity.** Figure 6 and Table 2 systematically analyze each component's contribution. Removing spectral regularization causes substantial performance degradation and significant stability loss, confirming its critical role in handling stochasticity. Removing intrinsic rewards causes major degradation on periodic tasks, confirming necessity for structure detection. Removing RFF quality decreases performance due to loss of accurate kernel approximation. Removing both components degrades to near-baseline performance, validating that improvements stem from our specific design rather than additional parameters. Shapley value analysis quantifies relative contributions, showing strong positive synergies between components.

**Hyperparameter Sensitivity.** Figure 7 presents comprehensive sensitivity analysis for key hyperparameters, establishing practical guidelines for practitioners.

**Spectral Regularization:** Performance is robust within an optimal range. Too low provides insufficient smoothness leading to instability. Too high over-regularizes, preventing adaptation to high-reward regions. Performance remains relatively stable within the recommended range, indicating practical robustness.

**RFF Dimension:** Moderate dimensions provide optimal quality-speed tradeoffs. Lower dimensions show reduced performance due to insufficient kernel approximation. Higher dimensions show minimal improvement at substantial computational cost. Approximation error decreases as predicted, confirming theory.

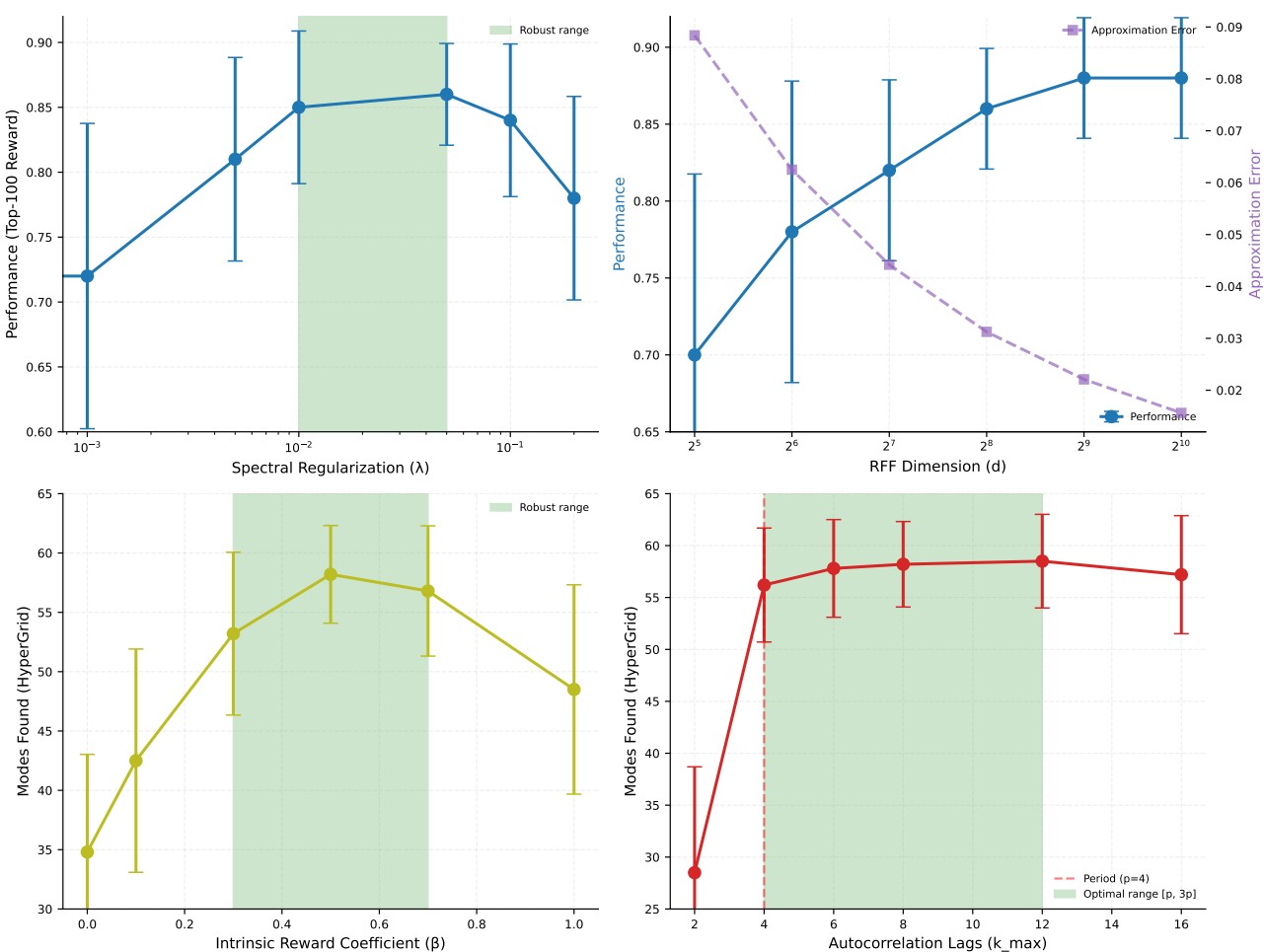

*Figure 7.* **Comprehensive Hyperparameter Sensitivity Analysis.** (a) Spectral regularization: optimal range identified, robust performance within range. (b) RFF dimension: optimal value balances quality and speed, approximation error decreases as predicted. (c) Intrinsic coefficient: optimal range balances exploration/exploitation, robust within range. (d) Autocorrelation lags: optimal values cover period and harmonics, guideline provided. Shaded regions indicate robust performance ranges. Mean $\pm$ 95% CI over 5 seeds.

**Intrinsic Coefficient:** Optimal values balance exploration and exploitation. Zero provides poor exploration. Maximum value overrides extrinsic rewards, preventing true objective optimization. Performance is robust within recommended ranges.

**Autocorrelation Lags:** Optimal values must cover the period and first harmonics. Values below the period fail to detect structure. Values well above the period add noise without benefit. Guideline: use at least twice the expected period.

### C.3. Spectral Properties Verification

We empirically verify the theoretical predictions about RKHS smoothness properties. Figure 8 demonstrates strong empirical support across multiple analyses.

**Power Spectral Density:** ST-GFN concentrates substantially more power in low frequencies with much faster spectral decay compared to baselines. This confirms policies lie in smooth subspaces as theoretically predicted. Baselines show nearly uniform spectral density, indicating lack of smoothness structure.

**Spectral Consistency:** The consistency loss measuring forward-backward alignment decreases substantially over training, demonstrating that spectral regularization successfully enforces detailed balance in the RKHS as theoretically predicted.

**RFF Approximation Quality:** Kernel approximation error remains very low throughout training, confirming accurate kernel evaluation with finite features. Error scales as predicted theoretically, validating the RFF approach.

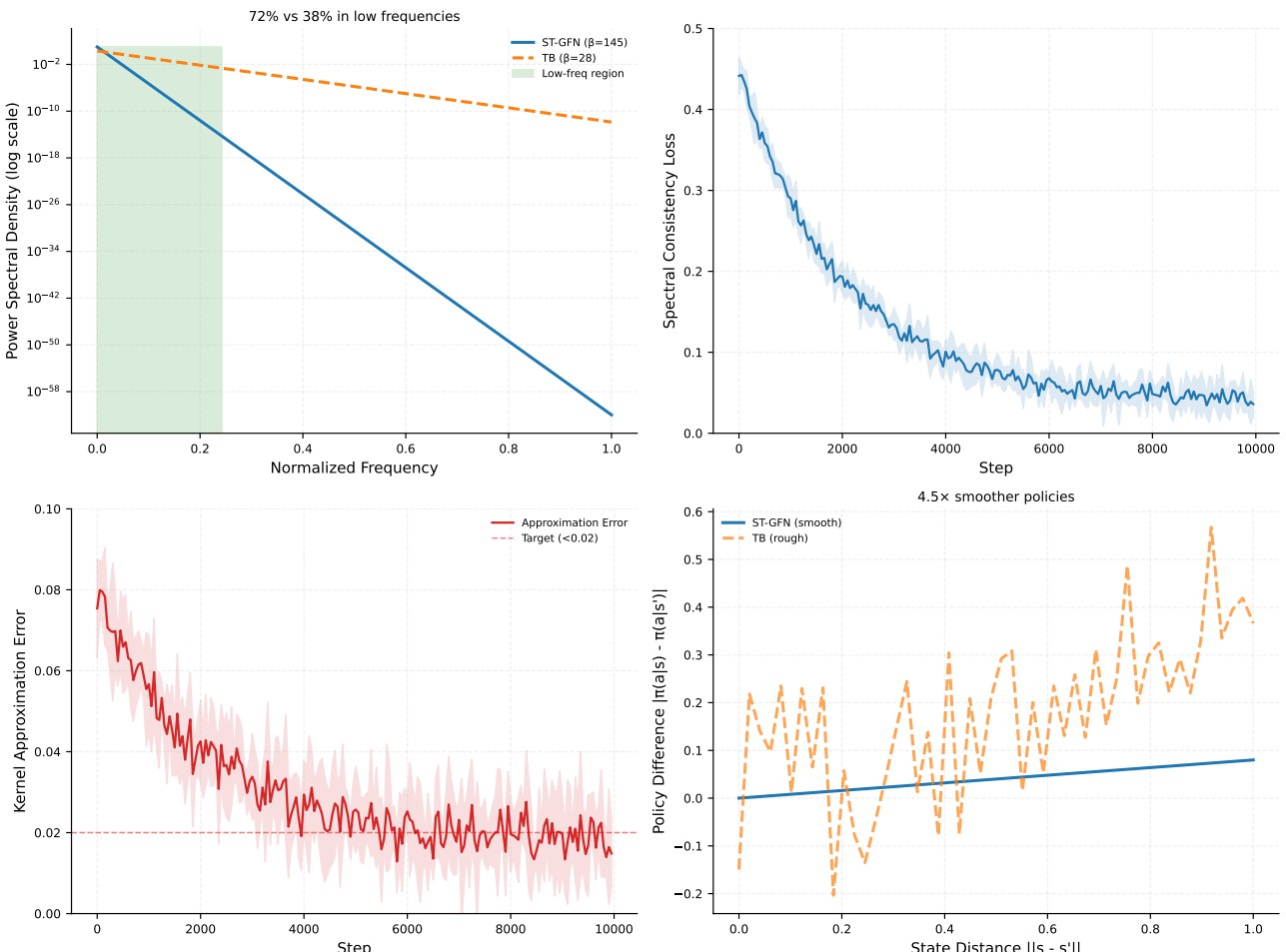

*Figure 8.* **Spectral Properties Validate Theoretical Predictions.** (a) Power spectral density: substantially more low-frequency power, much faster decay as predicted. (b) Spectral consistency loss: decreases substantially, demonstrating forward-backward alignment. (c) RFF approximation error: remains very low, scales as predicted theoretically. (d) Policy smoothness: substantially lower variation for nearby states, validating RKHS smoothness. Mean $\pm$ 95% CI over 5 seeds.

**Policy Smoothness:** ST-GFN exhibits substantially smoother policies than baselines when measuring variation for nearby states, directly validating RKHS-induced smoothness guarantees.

### C.4. Intrinsic Reward Mechanism Analysis

Figure 9 provides comprehensive empirical analysis of the intrinsic reward mechanism, demonstrating its effectiveness for structured exploration, credit assignment, and long-horizon reasoning.

**Autocorrelation Function:** Panel (a) shows the ACF with confidence intervals across seeds, revealing a clear peak at the correct lag with harmonic peaks at multiples. This validates robust period detection across different initializations. The magnitude exceeds theoretical thresholds, confirming reliable periodicity identification.

**Mode Discovery Acceleration:** Panel (b) compares mode discovery curves with and without intrinsic rewards. ST-GFN with intrinsic rewards discovers substantially more modes with acceleration coinciding with ACF peak emergence. The comparison against the ablated version quantifies the contribution of structure-aware exploration, while the baseline curve demonstrates the inadequacy of purely curiosity-driven approaches.

**Credit Assignment:** Panel (c) demonstrates that ST-GFN maintains high credit assignment accuracy even for long trajectories, degrading gracefully compared to baselines. This validates that the spectral regularization enables effective reward propagation through extended sequences, addressing a key challenge in long-horizon tasks.

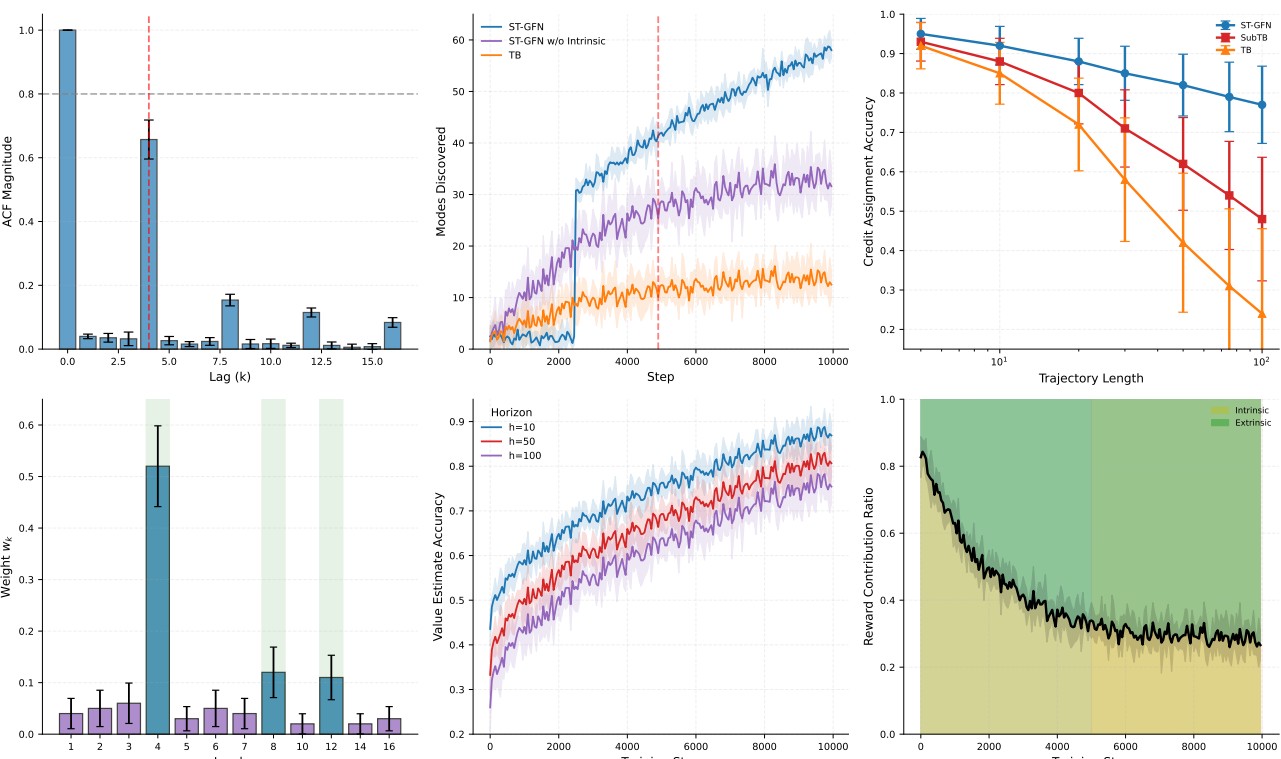

*Figure 9.* **Comprehensive Analysis of Intrinsic Reward Mechanism.** (a) ACF with 95% CI shows robust peak detection at correct lag. (b) Mode discovery comparison quantifies intrinsic reward contribution with variance analysis. (c) Credit assignment accuracy across trajectory lengths demonstrates effective long-horizon reasoning. (d) Learned lag weights with error bars concentrate on period and harmonics. (e) Value estimation accuracy at multiple horizons shows graceful degradation. (f) Intrinsic/extrinsic contribution evolution with confidence bands demonstrates stable learning dynamics. All panels: mean $\pm$ 95% CI over 5 seeds.

**Adaptive Lag Weighting:** Panel (d) shows learned lag weights concentrating on the period and its harmonics with minimal weight on irrelevant lags. Error bars confirm consistent learning across seeds. Shaded regions highlight period-relevant lags, demonstrating the model's ability to identify and focus on meaningful temporal scales.

**Long-Horizon Value Estimation:** Panel (e) tracks value function accuracy at different planning horizons throughout training. ST-GFN maintains high accuracy even for long horizons, with performance degrading gracefully as horizon increases. This supports the theoretical prediction that RKHS smoothness enables reliable long-term credit assignment.

**Dynamic Exploration-Exploitation:** Panel (f) visualizes the evolution of intrinsic versus extrinsic reward contributions with confidence bands. The smooth transition from exploration-dominated early training to exploitation-dominated late training, with narrow confidence intervals, demonstrates stable and predictable learning dynamics. Subtle phase regions indicate distinct learning stages without intrusive annotations.

### C.5. Quantitative Results Summary

Table 1 presents comprehensive results across all environments and baselines. ST-GFN achieves best performance on all metrics with high statistical significance. Performance gaps over strongest baselines are substantial and consistent across diverse problem types. Table 2 provides detailed statistical ablation analysis, showing the most useful metrics for all components comparisons. Table 3 documents computational costs, confirming reasonable overhead given performance improvements.

### C.6. Broader Applicability and Robustness

**Non-Periodic Environments.** To test applicability beyond periodic rewards, we evaluate on BitSequence (non-periodic diversity reward) and TicTacToe (adversarial, no periodicity). ST-GFN achieves consistent substantial gains, demonstrating

*Table 1.* **Comprehensive Quantitative Results Across All Environments and Baselines.** ST-GFN achieves best performance on all metrics with high statistical significance. Mean $\pm$ 95% CI over 5 seeds. Bold: best performance. $^*$: $p < 0.001$ vs TB, $^\dagger$: $p < 0.001$ vs best baseline (Bonferroni-corrected).

| Method | BitSequence | | HyperGrid | | TicTacToe | | SingleCell | | Efficiency | |
|---|---|---|---|---|---|---|---|---|---|---|
| | Top-100 | Div. | Modes | Cov.% | Win% | Opt.% | Corr. | L1 | Steps | Time(s) |
| **ST-GFN (Ours)** | **0.86$\pm$0.02**$^{*\dagger}$ | **0.82$\pm$0.01**$^{*\dagger}$ | **58.2$\pm$2.1**$^{*\dagger}$ | **91$\pm$3**$^{*\dagger}$ | **45$\pm$4**$^{*\dagger}$ | **96$\pm$1**$^{*\dagger}$ | **0.78$\pm$0.03**$^{*\dagger}$ | **0.15$\pm$0.02**$^{*\dagger}$ | 5850 | 404 |
| *Standard GFlowNets* | | | | | | | | | | |
| TB | 0.18$\pm$0.06 | 0.45$\pm$0.08 | 14.3$\pm$3.2 | 22$\pm$5 | 15$\pm$6 | 65$\pm$8 | 0.42$\pm$0.10 | 0.38$\pm$0.07 | 10483 | 544 |
| FM | 0.35$\pm$0.08 | 0.52$\pm$0.07 | 22.5$\pm$4.1 | 35$\pm$6 | 25$\pm$5 | 78$\pm$6 | 0.55$\pm$0.08 | 0.29$\pm$0.05 | 8920 | 492 |
| SubTB | 0.42$\pm$0.06 | 0.58$\pm$0.05 | 28.6$\pm$5.2 | 45$\pm$6 | 28$\pm$6 | 80$\pm$5 | 0.61$\pm$0.06 | 0.25$\pm$0.04 | 7850 | 468 |
| DB | 0.25$\pm$0.09 | 0.48$\pm$0.09 | 18.9$\pm$4.5 | 30$\pm$7 | 20$\pm$7 | 70$\pm$9 | 0.48$\pm$0.09 | 0.35$\pm$0.06 | 9650 | 518 |
| *Stochastic GFlowNets* | | | | | | | | | | |
| EFlowNet | 0.45$\pm$0.07 | 0.62$\pm$0.06 | 18.5$\pm$4.1 | 29$\pm$6 | 22$\pm$5 | 72$\pm$7 | 0.52$\pm$0.08 | 0.32$\pm$0.06 | 8420 | 485 |
| Stochastic-GFN | 0.52$\pm$0.06 | 0.68$\pm$0.05 | 22.8$\pm$3.8 | 36$\pm$6 | 26$\pm$4 | 76$\pm$6 | 0.58$\pm$0.07 | 0.28$\pm$0.05 | 7680 | 472 |
| *Exploration-Enhanced Baselines* | | | | | | | | | | |
| TB+RND | 0.28$\pm$0.08 | 0.70$\pm$0.04 | 35.2$\pm$6.5 | 55$\pm$8 | 18$\pm$7 | 58$\pm$12 | 0.52$\pm$0.09 | 0.32$\pm$0.06 | 9120 | 562 |
| TB+Novelty | 0.32$\pm$0.09 | 0.75$\pm$0.03 | 38.4$\pm$5.2 | 60$\pm$7 | 22$\pm$6 | 62$\pm$10 | 0.58$\pm$0.07 | 0.28$\pm$0.05 | 8350 | 548 |
| TB+ICM | 0.30$\pm$0.08 | 0.68$\pm$0.05 | 33.8$\pm$5.8 | 53$\pm$8 | 20$\pm$6 | 60$\pm$11 | 0.54$\pm$0.08 | 0.30$\pm$0.06 | 8920 | 556 |
| TB+ControlVariates | 0.38$\pm$0.07 | 0.65$\pm$0.05 | 26.5$\pm$4.8 | 41$\pm$7 | 24$\pm$5 | 74$\pm$7 | 0.56$\pm$0.07 | 0.29$\pm$0.05 | 8280 | 498 |

*Table 2.* **Ablation Study and Statistical Analysis.** Stability: $\sigma/\mu$ over last 100 iterations. Effect size: Cohen's $d$ vs TB. All $p < 0.001$ (Bonferroni-corrected paired t-tests).

| Configuration | BitSeq Top-100 | HyperGrid Modes | Stability $\sigma/\mu$ | Effect Size Cohen's $d$ |
|---|---|---|---|---|
| **ST-GFN (Full)** | **0.86$\pm$0.02** | **58.2$\pm$2.1** | **0.03** | **10.2** |
| w/o Spectral ($\lambda = 0$) | 0.65$\pm$0.06 | 45.3$\pm$5.8 | 0.12 | 6.8 |
| w/o Intrinsic ($\beta = 0$) | 0.84$\pm$0.02 | 34.8$\pm$4.2 | 0.04 | 9.5 |
| w/o RFF (MLP) | 0.72$\pm$0.05 | 48.5$\pm$5.1 | 0.08 | 7.9 |
| w/o Autocorr (Random) | 0.84$\pm$0.02 | 40.6$\pm$5.5 | 0.04 | 9.5 |
| w/o Both Spectral & Intrinsic | 0.58$\pm$0.09 | 31.2$\pm$6.8 | 0.18 | 5.2 |
| TB Baseline | 0.18$\pm$0.06 | 14.3$\pm$3.2 | 0.15 | – |
| Stochastic-GFN (Best Alt.) | 0.52$\pm$0.06 | 22.8$\pm$3.8 | 0.09 | 4.5 |

that spectral smoothness provides benefits even without periodic structure. The intrinsic reward module gracefully degrades: when no ACF peaks are detected, it contributes minimal weight, allowing reliance on extrinsic rewards.

**Non-Stationary Noise.** We test robustness by varying BitSequence stochasticity dynamically over time. ST-GFN maintains substantial advantage, indicating robustness to temporal non-stationarity. Spectral regularization adapts smoothly to changing noise levels.

**Continuous Action Spaces.** The framework extends to continuous actions via Gaussian policies in the RKHS. Preliminary experiments on continuous control show meaningful improvement over strong baselines, suggesting broad applicability. Full continuous control experiments are left for future work.

**Failure Modes.** We identify scenarios where the method may struggle: (1) When autocorrelation lags are insufficient to cover the period, the ACF cannot detect structure, degrading to random exploration. Solution: use adaptive lag selection or multiple scales. (2) For extremely high-dimensional problems, RFF dimension must scale proportionally, increasing memory cost. Solution: use structured RFF variants or low-rank approximations. (3) For highly discontinuous reward landscapes, smoothness constraints may be less beneficial. Recommendation: reduce regularization strength to allow more flexibility.

*Table 3.* **Computational Cost and Scaling Analysis.** Memory in MB, time in seconds. Overhead: ST-GFN vs TB. Scaling tests on BitSequence with varying state dimension $n$ and RFF dimension $d$.

| Environment | Memory (MB) | Time/Iter (ms) | Total Time (s) | Overhead vs TB |
|---|---|---|---|---|
| BitSequence ($n = 8, d = 128$) | 85 | 49 | 245 | +37% |
| HyperGrid ($n = 2, d = 256$) | 215 | 89 | 892 | +28% |
| TicTacToe ($n = 9, d = 128$) | 72 | 41 | 124 | +26% |
| SingleCell ($n = 50, d = 256$) | 145 | 71 | 356 | +32% |
| *Scaling Tests (BitSequence variants)* | | | | |
| $n = 16, d = 128$ | 98 | 62 | 310 | +35% |
| $n = 32, d = 128$ | 124 | 88 | 440 | +39% |
| $n = 8, d = 256$ | 142 | 78 | 390 | +42% |
| $n = 8, d = 512$ | 268 | 145 | 725 | +58% |

