# Spectral Flow Matching: Stabilizing Stochastic GFlowNets via Frequency-Domain Regularization

January 29, 2026

## Abstract

Generative Flow Networks (GFNs) offer a powerful paradigm for diverse sampling, yet they often exhibit instability and poor convergence when applied to stochastic or sparse-reward environments. To mitigate the high variance inherent in these settings, we propose a fundamental re-framing of the GFlowNet training objective within the frequency domain. We present **Spectral Time-Dependent GFlowNets (ST-GFNs)**, a framework that leverages Fourier analysis to enforce smoothness and stability in learned policies. Our theoretical analysis proves that our proposed spectral loss is mathematically equivalent to regularized value iteration, acting as a principled low-pass filter that separates signal from noise. Furthermore, we tackle the challenge of exploration in sparse landscapes by introducing a novel autocorrelated intrinsic reward derived from the Wiener-Khinchin theorem. Through extensive experiments ranging from adversarial games and noisy sequence generation to high-dimensional single-cell perturbation modeling, we demonstrate that ST-GFNs significantly outperform existing baselines in terms of robustness, sample efficiency, and mode discovery.[1]

## 1. Introduction

Generative Flow Networks (GFNs) (Bengio et al., 2021b) excel at sampling diverse, high-quality candidates but are severely challenged by stochastic environments, where they often suffer from high-variance credit assignment, unstable training, and poor generalization (Jiralerspong et al., 2024; Pan et al., 2023; Malkin et al., 2022). To address these fundamental issues, we propose a paradigm shift: recasting the entire GFlowNet problem in the frequency domain. We introduce **Spectral Time-Dependent GFlowNets (ST-GFNs)**, a novel framework built upon a unified spectral loss

---

[1]To ensure anonymity during the review process, the code will be made publicly available via a GitHub repository immediately following the rebuttal period.

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

| *Standard GFlowNets* | | | | | | | | | | |
| TB | 0.18±0.06 | 0.45±0.08 | 14.3±3.2 | 22±5 | 15±6 | 65±8 | 0.42±0.10 | 0.38±0.07 | 10483 | 544 |
| FM | 0.35±0.08 | 0.52±0.07 | 22.5±4.1 | 35±6 | 25±5 | 78±6 | 0.55±0.08 | 0.29±0.05 | 8920 | 492 |
| SubTB | 0.42±0.06 | 0.58±0.05 | 28.6±5.2 | 45±6 | 28±6 | 80±5 | 0.61±0.06 | 0.25±0.04 | 7850 | 468 |
| DB | 0.25±0.09 | 0.48±0.09 | 18.9±4.5 | 30±7 | 20±7 | 70±9 | 0.48±0.09 | 0.35±0.06 | 9650 | 518 |
| *Stochastic GFlowNets* | | | | | | | | | | |
| EFlowNet | 0.45±0.07 | 0.62±0.06 | 18.5±4.1 | 29±6 | 22±5 | 72±7 | 0.52±0.08 | 0.32±0.06 | 8420 | 485 |
| Stochastic-GFN | 0.52±0.06 | 0.68±0.05 | 22.8±3.8 | 36±6 | 26±4 | 76±6 | 0.58±0.07 | 0.28±0.05 | 7680 | 472 |
| *Exploration-Enhanced Baselines* | | | | | | | | | | |
| TB+RND | 0.28±0.08 | 0.70±0.04 | 35.2±6.5 | 55±8 | 18±7 | 58±12 | 0.52±0.09 | 0.32±0.06 | 9120 | 562 |
| TB+Novelty | 0.32±0.09 | 0.75±0.03 | 38.4±5.2 | 60±7 | 22±6 | 62±10 | 0.58±0.07 | 0.28±0.05 | 8350 | 548 |
| TB+ICM | 0.30±0.08 | 0.68±0.05 | 33.8±5.8 | 53±8 | 20±6 | 60±11 | 0.54±0.08 | 0.30±0.06 | 8920 | 556 |
| TB+ControlVariates | 0.38±0.07 | 0.65±0.05 | 26.5±4.8 | 41±7 | 24±5 | 74±7 | 0.56±0.07 | 0.29±0.05 | 8280 | 498 |

*Table 2.* **Ablation Study and Statistical Analysis.** Stability: $\sigma/\mu$ over last 100 iterations. Effect size: Cohen's $d$ vs TB. All $p < 0.001$ (Bonferroni-corrected paired t-tests).

| Configuration | BitSeq | HyperGrid | Stability | Effect Size |
| --- | --- | --- | --- | --- |
| | Top-100 | Modes | $\sigma/\mu$ | Cohen's $d$ |
| **ST-GFN (Full)** | **0.86±0.02** | **58.2±2.1** | **0.03** | **10.2** |
| w/o Spectral ($\lambda = 0$) | 0.65±0.06 | 45.3±5.8 | 0.12 | 6.8 |
| w/o Intrinsic ($\beta = 0$) | 0.84±0.02 | 34.8±4.2 | 0.04 | 9.5 |
| w/o RFF (MLP) | 0.72±0.05 | 48.5±5.1 | 0.08 | 7.9 |
| w/o Autocorr (Random) | 0.84±0.02 | 40.6±5.5 | 0.04 | 9.5 |
| w/o Both Spectral & Intrinsic | 0.58±0.09 | 31.2±6.8 | 0.18 | 5.2 |
| TB Baseline | 0.18±0.06 | 14.3±3.2 | 0.15 | – |
| Stochastic-GFN (Best Alt.) | 0.52±0.06 | 22.8±3.8 | 0.09 | 4.5 |

that spectral smoothness provides benefits even without periodic structure. The intrinsic reward module gracefully degrades: when no ACF peaks are detected, it contributes minimal weight, allowing reliance on extrinsic rewards.

**Non-Stationary Noise.** We test robustness by varying BitSequence stochasticity dynamically over time. ST-GFN maintains substantial advantage, indicating robustness to temporal non-stationarity. Spectral regularization adapts smoothly to changing noise levels.

**Continuous Action Spaces.** The framework extends to continuous actions via Gaussian policies in the RKHS. Preliminary experiments on continuous control show meaningful improvement over strong baselines, suggesting broad applicability. Full continuous control experiments are left for future work.

**Failure Modes.** We identify scenarios where the method may struggle: (1) When autocorrelation lags are insufficient to cover the period, the ACF cannot detect structure, degrading to random exploration. Solution: use adaptive lag selection or multiple scales. (2) For extremely high-dimensional problems, RFF dimension must scale proportionally, increasing memory cost. Solution: use structured RFF variants or low-rank approximations. (3) For highly discontinuous reward landscapes, smoothness constraints may be less beneficial. Recommendation: reduce regularization strength to allow more flexibility.

*Table 3.* **Computational Cost and Scaling Analysis.** Memory in MB, time in seconds. Overhead: ST-GFN vs TB. Scaling tests on BitSequence with varying state dimension $n$ and RFF dimension $d$.

| Environment | Memory (MB) | Time/Iter (ms) | Total Time (s) | Overhead vs TB |
|---|---|---|---|---|
| BitSequence ($n = 8, d = 128$) | 85 | 49 | 245 | +37% |
| HyperGrid ($n = 2, d = 256$) | 215 | 89 | 892 | +28% |
| TicTacToe ($n = 9, d = 128$) | 72 | 41 | 124 | +26% |
| SingleCell ($n = 50, d = 256$) | 145 | 71 | 356 | +32% |
| *Scaling Tests (BitSequence variants)* | | | | |
| $n = 16, d = 128$ | 98 | 62 | 310 | +35% |
| $n = 32, d = 128$ | 124 | 88 | 440 | +39% |
| $n = 8, d = 256$ | 142 | 78 | 390 | +42% |
| $n = 8, d = 512$ | 268 | 145 | 725 | +58% |