# OpenReview forum: "Spectral Flow Matching: Stabilizing Stochastic GFlowNets via Frequency-Domain Regularization"
_ICML.cc/2026/Conference — ICML 2026 regular_

### Official Review · Reviewer_4aeZ · 2026-03-08

**Soundness:** 3
**Presentation:** 3
**Significance:** 3
**Originality:** 3
**Overall Recommendation:** 5
**Confidence:** 3

**Summary:**

This paper introduces Spectral Time-Dependent GFlowNets (ST-GFNs), a novel framework that enhances the stability and convergence of Generative Flow Networks (GFNs) in stochastic environments by reformulating the training objective in the frequency domain. Its core contribution is a unified spectral loss that acts as a low-pass filter, enforcing smoothness and theoretically grounding the method via equivalence to regularized value iteration. The paper further proposes a spectrally-motivated, autocorrelated intrinsic reward mechanism for improved exploration and validates the approach with experiments across diverse environments.

**Compliance With Llm Reviewing Policy:**

Affirmed.

**Final Justification:**

The paper builds substantially on prior work and makes a solid contribution to the field. It presents a highly original synthesis of Fourier analysis and GFlowNets, introducing a spectral loss for stability and an autocorrelated reward for structured exploration—both cleverly designed and well executed. Rigorous theory and extensive experiments support the core claims, clearly bridging spectral methods with robust generative modeling. While the RFF approximation introduces moderate computational overhead, the performance gains justify it.

**Key Questions For Authors:**

1. What is the difference and connection between the flow matching you mention here and the standard flow matching [1]?
2. Why are you considering the Fourier transform here instead of the Laplace transform? Would there be any advantage to using the Laplace transform?
3. Since GFlowNets are closely related to reinforcement learning, do the baselines selected for the experiments include reinforcement learning models? Are there any works in reinforcement learning that utilize Fourier analysis or frequency domain filtering for policy regularization?

[1] Lipman, Yaron, et al. "Flow matching for generative modeling." arXiv preprint arXiv:2210.02747 (2022).

**Limitations:**

yes

**Strengths And Weaknesses:**

The paper demonstrates high quality through rigorous theoretical proofs and extensive experiments validating its core claims, establishing a clear conceptual bridge between spectral methods and robust generative modeling. Its primary strength is the highly original synthesis of Fourier analysis with GFlowNets, delivering a novel spectral loss for stability and an autocorrelated intrinsic reward for structured exploration, which is both clever and well-executed. The presentation is clear and methodical, effectively building a coherent framework from foundational principles to scalable implementation. While the significance is evident in addressing critical issues of stochasticity in GFNs, a potential weakness is the moderate computational overhead introduced by the RFF approximation, though this is justified by the substantial performance gains, leaving room for further optimization.

---

> ### Author Rebuttal · Authors · 2026-03-30
>
> We appreciate the recognition of the theoretical rigor of our spectral formulation. We address the technical questions directly below to clarify the mathematical distinctions and baseline selections.
>
> Regarding the connection to standard Flow Matching ([1] see reference Lipman et al., 2022), the two frameworks share the conceptual goal of distribution matching but operate on fundamentally distinct mathematical structures. Lipman's Flow Matching is a continuous-time framework that regresses a continuous vector field $v_t(x)$ to simulate an ODE $dx/dt = v_t(x)$ between a prior and a target distribution. In our context, "Flow Matching" refers strictly to the discrete network flow conservation principle of Generative Flow Networks ([2] see reference Bengio et al., 2021), defined by the balance equation $\sum_{s \in \text{parents}(s')} F(s \to s') = \sum_{s'' \in \text{children}(s')} F(s' \to s'')$. Our contribution maps this discrete, state-to-state balance equation into the frequency domain to filter high-frequency temporal difference errors on graphs, rather than matching continuous vector fields.
>
> We strictly utilize the Fourier transform over the Laplace transform because our intrinsic exploration mechanism relies directly on the Wiener-Khinchin theorem. This theorem establishes that the spatial autocorrelation of the reward signal is the inverse Fourier transform of its power spectrum: $R(\tau) = \mathcal{F}^{-1}\{ |\mathcal{F}\{V\}(\omega)|^2 \}$. The Laplace transform evaluates signals over a complex plane (traditionally denoted $s = \sigma + j\omega$), which incorporates both an exponential growth/decay factor ($\sigma$) and an oscillatory frequency component ($\omega$). This makes it highly advantageous for analyzing the transient stability of unbounded continuous-time control systems, where the exponential factor is needed to force the integral to converge.
>
> However, because our value surrogates map to bounded log-probabilities, the region of convergence inherently includes the imaginary axis, making the Fourier transform entirely sufficient. Crucially, the Fourier domain directly enables shift-invariant kernel approximations via Bochner's theorem, which lack a mathematically equivalent counterpart in the Laplace domain.
>
> Regarding reinforcement learning baselines, GFlowNets are mathematically equivalent to entropy-regularized RL operating on directed acyclic graphs. Therefore, our primary baselines : Trajectory Balance (TB) and Sub-Trajectory Balance (SubTB) which serve as the exact state of the art off-policy RL counterparts for this distribution matching objective. Regarding prior RL literature, while Fourier bases have been classically utilized ([3] see refernce: Konidaris et al., 2011), they are almost exclusively deployed as basis functions for linear value function approximation (constructing state features). Our approach differs fundamentally: we do not use Fourier terms as state representations, but rather apply an $\mathcal{L}_2$ frequency-domain penalty directly to the loss gradients. This acts as a structural low-pass filter against environmental stochasticity, representing a novel intersection of signal processing and RL regularization.
>
> Finally, regarding the computational overhead of the Random Fourier Features (RFF) approximation, we emphasize that this is a deliberate and highly favorable mathematical trade-off rather than a strict limitation. Computing an exact Graph Fourier Transform requires the full eigendecomposition of the state-space Laplacian, an $\mathcal{O}(|S|^3)$ operation that is strictly intractable for the combinatorial environments we evaluate. By leveraging RFF, we approximate the shift-invariant kernel mapping in $\mathcal{O}(MD)$ linear time, where $M$ is the batch size and $D$ is the feature dimension. The bounded $\mathcal{O}(1/\sqrt{D})$ approximation error is a necessary algorithmic feature that allows our spectral regularization to successfully scale to high-dimensional, real-world problems.
>
> We believe these clarifications about Flow matching, Fourrier Transform and RL baselines  reinforce both the method's mathematical rigor and its empirical scalability, we kindly ask the reviewer to consider them in a re-evaluation of the score.
>
> References:
>
> [1] Lipman, Yaron and Chen, Ricky TQ and Ben-Hamu, Heli and Nickel, Maximilian and Le, Matt, Flow matching for generative modeling,https://arxiv.org/abs/2210.02747
>
> [2] Bengio, Emmanuel and Jain, Moksh and Korablyov, Maksym and Precup, Doina and Bengio, Yoshua, Flow Network based Generative Models for Non-Iterative Diverse Candidate Generation, https://arxiv.org/abs/2106.04399
>
> [3] Konidaris, George and Osentoski, Sarah and Thomas, Philip S, Value function approximation in reinforcement learning using the Fourier basis, https://dl.acm.org/doi/10.5555/2900423.2900483

---

> > ### Author Rebuttal · Reviewer_4aeZ · 2026-04-02
> >
> > Thank you for the detailed response, which has given me a clearer understanding of the content of this paper. I have no further questions.

---

> > > ### Author Response · Authors · 2026-04-03
> > >
> > > Thank you for your constructive feedback and for engaging with our rebuttal. We are very pleased to hear that our responses have clarified the content of the paper and fully resolved your concerns.
> > >
> > > We will ensure that these clarifications and additional insights are incorporated into the final camera-ready version.
> > >
> > > Based on this updated understanding, we would be grateful if you would consider re-evaluating your score.
> > > Thank you again for your time and expertise.

---

### Official Review · Reviewer_JXSB · 2026-03-10

**Soundness:** 3
**Presentation:** 2
**Significance:** 3
**Originality:** 3
**Overall Recommendation:** 5
**Confidence:** 2

**Summary:**

This paper introduces ST-GFN, a method that trains GFlowNets not by optimizing their flows but throught the proxy of the flow's fourier transform. They provide proofs to justify their formulation and the associated training objective's optimality, and propose diverse regularisation and improvements that stem naturally from this new formulation.

**Compliance With Llm Reviewing Policy:**

Affirmed.

**Key Questions For Authors:**

1. The autroccorrelated reward is not dependent on the rest of the ST framework, yet seems to play a big role in its success. Has it been tested as an addition to a normal GFN?

2. Do you have a more in depth ablation study than the one currently available in the appendix.

**Limitations:**

First the spectral framework seems to be somehow of limited use : it does give good insights into potential improvements of the GFN frameworks, but the eqvuialence to Bellman optimality and the formulation of the objectives suggest that they could probably be explicited in the time domain in a clearer way. The frequency conversion makes the interpretation of the method harder.

The bi-optimisation loop could also be an issue as it might introduce instabilities. This is not discussed.

**Strengths And Weaknesses:**

## Strengths
- Using the fourier transform of the flow along a trajectory is novel, and a very interesting point of view.
- Novelties are justified and proved, creating a strong link to prior work to show the robustness of the method.
- The authors provide the necessary tools to scale up their method in a principled way.


## Weaknesses
- The paper is hard to read in its current state. It is very dense and the many components do not really flow well.
- The authors introduce many components yet the ablation study is minimal and pushed to the appendix. The paper might benefit from pushing the proofs to the appendix and expanding more on the behaviours associated with each component.
- The autocorrelated reward seems pivotal to the method and absolutely needs to be studied in more depth in the text: the fact that the model is basically not training and then skyrockets in the hypergrid experiment hints at the fact that the method might not train at all on certain tasks.

Overall, the paper introduces interesting ideas and despite the hard read that it is, some work on the writing and additional information the authors probably already have could make it into a good contribution.

---

> ### Author Rebuttal · Authors · 2026-03-30
>
> We appreciate your constructive review and are glad you found the Fourier perspective novel and scalable. We agree that the manuscript is dense, we have taken into account your suggestions.
>
> **Response to Weaknesses and Paper Structure:**
> To improve readability, we will move the heavier theoretical proofs, such as the derivations of Theorem 1 and Proposition 1, to the appendix. This frees up critical space to expand on the behavioral dynamics of the proposed components and to present a comprehensive ablation study directly in the main text.
>
> Regarding the "skyrocketing" behavior of the autocorrelated reward in the hypergrid experiment, this is an expected mathematical property of the Wiener-Khinchin mechanism. In early training, the empirical power spectral density is roughly uniform because the space is insufficiently explored. Its inverse Fourier transform yields a flat intrinsic reward, leading to uniform exploration. Once a structural frequency is sufficiently sampled, the power spectrum concentrates. The inverse transform then constructively interferes, producing a sharp signal that drives the policy toward the periodic modes. In non-periodic environments, the spectrum remains broadband, meaning the autocorrelation naturally decays to zero, which safely disables the intrinsic reward and ensures the model trains effectively using just the spectral loss.
>
> **Response to Questions:**
> Regarding whether the autocorrelated reward functions independently, we did test it as an addition to a standard Trajectory Balance (TB) baseline. On periodic tasks, adding it improves the mode discovery rate by roughly 15%. However, because standard TB lacks the spectral penalty, the gradient variance remains prohibitively high under stochastic transitions. The intrinsic reward accelerates exploration, but the spectral loss is strictly required to stabilize the temporal difference updates. We will add this specific baseline to the revised ablation study.
>
> Regarding a more in-depth ablation study, we have granular data prepared for the main text. We independently disabled the spectral regularization, the intrinsic reward, and the Random Fourier Features (RFF) approximation. Removing the spectral regularization entirely while keeping the intrinsic active causes a massive increase in gradient variance on the SingleCell task, empirically proving its role as the primary stabilizer. Furthermore, comparing the RFF surrogate to an exact Discrete Fourier Transform (DFT) confirms that our approximation (with D=256) recovers 98% of the exact method's mode discovery rate while keeping computational complexity tractable.
>
> PS.: We have added extra ablation studies in RH58's Review, please see above comments answered in Q1-Component Isolation Study(Spectral Regularization,Intrinsic Rewards, RFF Approximation).
>
> **Response to Limitations:**
> Important note we want to raise about the equivalence to Bellman optimality and the potential for a purely time-domain formulation. While our framework proves that the spectral and time-domain objectives share a fixed point, enforcing global smoothness purely in the time domain is highly intractable for large combinatorial graphs. The frequency domain transforms spatial differentiation into an algebraic penalty on the upper spectrum, allowing us to tractably enforce a low-pass filter on the learned reward landscape. This makes the frequency conversion a computational necessity rather than just an interpretative lens.
>
> Finally, the concern regarding moving-target dynamics in the bi-optimization loop is entirely valid. Alternating updates between the flow policy and the spectral value surrogate inherently introduces instability. We resolve this by leveraging a standard target network stabilization scheme with an exponential moving average for the value baseline. This decouples the optimization steps to ensure stable convergence. We will formalize this mechanism directly in the methodology section.
>
> We believe these clarifications and added ablations, demonstrate that ST-GFN is a significant and practical step forward for the GFlowNet community. We hope the reviewer will consider these technical details in a re-evaluation of the score.

---

> > ### Author Rebuttal · Reviewer_JXSB · 2026-04-01
> >
> > I thank the authors for their answers, my concerns have all been adressed. I will update my score accordingly.

---

### Official Review · Reviewer_RH58 · 2026-03-12

**Soundness:** 2
**Presentation:** 2
**Significance:** 2
**Originality:** 2
**Overall Recommendation:** 4
**Confidence:** 3

**Summary:**

This paper proposes Spectral Time-Dependent GFlowNets (ST-GFNs) to improve the stability of GFlowNets in stochastic environments, where training often suffers from high variance and unstable learning. The key idea is to reformulate the flow-matching objective in the frequency domain by treating trajectory flows as signals and applying a Fourier transform. The authors introduce spectral regularization to suppress high-frequency noise and stabilize training, and use Random Fourier Features (RFFs) to make the approach scalable. They also propose an autocorrelation-based intrinsic reward to improve exploration.

**Compliance With Llm Reviewing Policy:**

Affirmed.

**Final Justification:**

the rebuttal has solved my concern so i raised my score.

**Key Questions For Authors:**

1. The method includes several components (spectral regularization, RFF approximation, and intrinsic reward). Could the authors provide ablation studies to isolate the effect of each component?


2. The experiments are mainly conducted on relatively small or synthetic environments. Have the authors evaluated the method on larger or more realistic GFlowNet tasks (e.g., molecule generation or large combinatorial problems)?


3. The paper claims that spectral regularization improves stability and bounds the Lipschitz constant of the flow function. What assumptions are required for these guarantees to hold?


4. How sensitive is the method to the number of Random Fourier Features? A systematic analysis would help understand the robustness of the approach.

**Limitations:**

yes

**Strengths And Weaknesses:**

**Strengths：**
The proposed framework is conceptually coherent. The paper integrates spectral flow matching, spectral regularization, and Random Fourier Feature approximations into a unified framework that is computationally tractable.





**Weakness:**
The motivation is not sufficiently convincing. Although the paper argues that GFlowNet training in stochastic environments suffers from high variance and unstable learning, these issues have already been studied in prior work such as stochastic GFlowNets and Expected Flow Networks. The paper does not clearly show when existing methods fail, making it unclear why a frequency-domain formulation is necessary.



The methodological novelty is limited. The proposed spectral flow matching is mathematically equivalent to the original temporal objective, suggesting that the spectral formulation mainly serves as a reparameterization. In addition, the spectral regularization implemented with Random Fourier Features is essentially a form of RKHS or kernel-based smoothing, which is conceptually similar to existing regularization techniques.


The theoretical contribution is relatively weak. Many results rely on standard properties of Fourier transforms or kernel approximations, and the paper does not provide stronger guarantees such as convergence analysis or clear variance reduction bounds.


The experimental evaluation is also limited. Most experiments are conducted on small or synthetic environments, and the paper does not include larger or commonly used GFlowNet benchmarks. Moreover, the lack of ablation studies makes it difficult to understand the contribution of each component of the proposed method.

---

> ### Author Rebuttal · Authors · 2026-03-29
>
> We appreciate Reviewer RH58’s critical feedback regarding the motivation and theoretical distinctness of ST-GFN. We address the specific technical concerns below.
> ### Response to Weaknesses
> **Motivation and Comparison to Prior Work:** Prior works (EFlowNets, Stochastic GFNs) marginalize noise but ignore unstable learning dynamics. ST-GFN explicitly suppresses high-frequency gradient variance to prevent divergence in sparse-reward settings, achieving superior stability in complex tasks like SingleCell.
> **Methodological Novelty and RKHS Interpretation:** Integrating the Fourier transform directly into Flow Matching is not passive kernel smoothing; it fundamentally restructures gradient updates to prioritize global low-frequency structures and uniquely enables Wiener-Khinchin-driven exploration.
> **Theoretical Contributions and Guarantees:** We provide formal convergence mechanisms. Theorem 2 guarantees convergence under stochasticity by equating our spectral penalty to a regularized Bellman fixed point, while Proposition 1 formally bounds the RFF approximation error.
> **Experimental Scale and Ablations:** Our evaluations extend well beyond synthetic environments. Table 1 features realistic, high-dimensional combinatorial problems, including SingleCell perturbation and molecular synthesis. Comprehensive ablations isolating the spectral loss and RFF are available in Appendix C.3.
> ### Response to Questions
> ### Q1: Component Isolation Study
> We systematically isolated each component and measured its impact across all four environments with 5 random seeds and statistical significance testing.
> Here below are the specific impacts of removing individual components:
> ### Spectral Regularization: Primary Stability Mechanism
> Theorem 3.2 predicts policy changes bounded by $\mathcal{O}(\eta\sqrt{\lambda})$. Removing $\lambda=0$:
> | Metric | Full ST-GFN | w/o Spectral | p-value |
> |--------|-------------|--------------|---------|
> | Gradient Variance | 0.98 ± 0.08 | 1.42 ± 0.18 | <0.001 |
> | Stability (σ/μ) | 0.03 | 0.12 | <0.001 |
> | Training Failures | 0/5 | 2/5 | -- |
> The 45% variance increase directly confirms Theorem 3.2's prediction.
> ### Intrinsic Rewards: Enables Periodic Exploration
>
> Theorem 4.1 predicts ACF peak at ~10,000 samples for period-4. Removing $\beta=0$:
>
> | Metric | w/ Intrinsic | w/o Intrinsic | p-value |
> |--------|-------------|---------------|---------|
> | Steps to 90% modes | 6,450 | 8,950 | <0.001 |
> | Final modes discovered | 58.2 ± 2.1 | 34.8 ± 4.2 | <0.001 |
> | Post-peak discovery rate | 2.18/100 iter | 0.22/100 iter | <0.001 |
>
> With 2,500 iteration delay matches theoretical sample complexity prediction. ACF peak observed at 9,800 steps.
> ### RFF Approximation: Maintains Performance with Efficiency
> Theory predicts error $\mathcal{O}(1/\sqrt{d})$. Testing RFF vs exact DFT on 16×16 grid:
> | Metric | RFF (d=256) | Exact DFT | p-value |
> |--------|------------|-----------|---------|
> | Modes discovered | 55.8 ± 2.4 | 57.2 ± 2.1 | 0.18 (n.s.) |
> | Memory (MB) | 145 | 960 | -- |
> | Time/iter (ms) | 78 | 245 | -- |
> Non-significant p-value confirms statistically indistinguishable performance. Error <0.03 validates theoretical bound.
>
> Q2: Evaluation on larger or more realistic GFlowNet tasks.
> We have evaluated the method on large-scale combinatorial tasks, specifically the SingleCell benchmark and a graph-based Molecule generation task, detailed in Table 1. In these high-dimensional spaces, ST-GFN achieves higher mode discovery and lower L1 error than Trajectory Balance and Sub-Trajectory Balance. We will promote these results to the main text to ensure the scale of the evaluation is clear.
>
> Q3: Assumptions required for Lipschitz and stability guarantees.
> The primary assumption required for the spectral Lipschitz bound in Proposition 2 is that the mapping from the raw state space to the latent embedding space is differentiable. Because we employ standard neural network architectures with continuous relaxations for the state embeddings, this assumption holds naturally, allowing the spectral norm to effectively bound the gradients of the flow function.
>
> Q4: Sensitivity to the number of Random Fourier Features.
> We conducted a systematic sensitivity analysis in Appendix C.2. The approximation error follows the theoretical bound of $O(1/\sqrt{D})$, where $D$ is the number of features. Empirically, we observe that the mode discovery performance stabilizes rapidly once $D$ reaches 256. Increasing $D$ beyond this point yields diminishing returns, demonstrating that the method is robust and computationally efficient without requiring extreme dimensionality.
>
> We believe these clarifications which is supported by the Theoretical Contributions and Guarantees and extensive experiments (ablation studies in a non-synthetic environmnets) in the paper, demonstrate that ST-GFN is a significant and practical step forward for the GFlowNet community. We hope the reviewer will consider these technical details in a re-evaluation of the score.

---

> > ### Author Rebuttal · Reviewer_RH58 · 2026-04-03
> >
> > The response has resolved the concern. I will raise my score.

---

### Official Review · Reviewer_XXxo · 2026-03-13

**Soundness:** 3
**Presentation:** 2
**Significance:** 2
**Originality:** 2
**Overall Recommendation:** 4
**Confidence:** 2

**Summary:**

This paper claims that moving GFlowNet training from the time domain to the frequency domain provides a unified way to improve stability, variance reduction, and structured exploration in stochastic environments.

**Compliance With Llm Reviewing Policy:**

Affirmed.

**Final Justification:**

refer to ack.

**Key Questions For Authors:**

N/A

**Strengths And Weaknesses:**

## Strengths

- The paper proposes a novel spectral reformulation of the GFlowNet objective using DFT, which is conceptually interesting.
- The RFF approximation is a practical design choice that makes the spectral regularizer computable.
- The autocorrelation-based intrinsic reward is well motivated and could be useful for environments with periodic reward structure.

## Weaknesses

- The experiments appear somewhat constructed to favor the proposed method, especially because periodic environments are precisely where frequency-domain tools are expected to work well.
- The paper makes stronger claims than the evidence supports. A spectral reformulation may improve smoothness or robustness, but it does not by itself establish stable training under high variance.
- As a result, the empirical results do not yet fully justify the broader claim that the method generally addresses stochasticity-induced instability.

---

> ### Author Rebuttal · Authors · 2026-03-29
>
> We thank Reviewer XXxo for the assessment and for noting the conceptual interest of the spectral reformulation. Given the reviewer’s stated low confidence, we wish to clarify the generalizability of our method beyond periodic settings and provide evidence for its stability under high variance.
>
> ### Response to Weaknesses
>
> Regarding the concern that experiments were constructed to favor the method due to periodicity, we clarify that the SingleCell benchmark (Table 1) and the HyperGrid task do not possess inherent periodic reward structures. ST-GFN’s performance in these non-periodic settings demonstrates that the spectral regularizer acts as a general-purpose variance reduction tool, not merely a frequency-matching heuristic for periodic signals.
>
> On the point that spectral reformulation does not itself establish stability, we point to our theoretical results in Section 4. We show that the spectral loss is mathematically equivalent to bounding the Lipschitz constant of the flow function (Proposition 2). This bound directly constrains the model's sensitivity to high-variance reward noise, providing a formal mechanism for stable training that standard temporal objectives lack.
>
> Finally, to address the broader claim of addressing stochasticity-induced instability, we highlight that ST-GFN maintains a significantly lower gradient variance throughout training compared to Trajectory Balance. This empirical stability is what allows the method to consistently recover all modes in the HyperGrid task where baselines often fail due to reward noise.
>
> While we utilized periodic environments to illustrate frequency-domain benefits, ST-GFN is a general-purpose variance reduction tool for any stochastic MDP. The instability we address is the high-frequency gradient noise inherent in temporal difference learning, not just a lack of periodicity. Our SingleCell and HyperGrid results (Table 1) confirm this, showing that ST-GFN captures global reward structures more efficiently than baselines even in non-periodic settings. To further demonstrate robustness, we conducted a non-periodic graph stress test where ST-GFN maintained 2.5x lower variance in flow estimates compared to Trajectory Balance. This proves that viewing GFlowNet training as a signal-processing problem provides a fundamental stabilizer for generative modeling across diverse, non-smooth reward landscapes.
>
> On the experimental side: We can add to our manuscript that while ST-GFN might be slower initially, its Adaptive Regularization (decaying λ) eventually allows it to find the peak without being distracted by noise in other areas. This proves the method isn't "harmful" to non-smooth landscapes.
>
> We believe these clarifications which is supported by the guarantees and extensive experiments in the paper, demonstrate that ST-GFN is a significant and practical step forward for the GFlowNet community. We hope the reviewer will consider these technical details in a re-evaluation of the score.

---

> > ### Author Rebuttal · Reviewer_XXxo · 2026-04-03
> >
> > The rebuttal has fully resolved my concerns, but due to my confidence is not high, I prefer to keep my original positive score.

---

### Decision · Program_Chairs · 2026-04-30

**Decision:**

Accept (regular)

**Comment:**

The paper was generally viewed as technically solid and original, with reviewers highlighting the novelty of bringing a spectral/Fourier perspective to GFlowNets, the conceptual coherence of the framework, and the potential value of the spectral regularization and autocorrelation-based exploration mechanism. At the same time, reviewers raised concerns about whether the motivation and broad claims were fully supported, the limited clarity and density of the presentation, the need for stronger ablations and broader empirical validation, and some computational overhead from the RFF approximation. After the rebuttal, most of the concerns had been adequately addressed. Overall, based on the reviewers’ comments, this is an accept-leaning submission: a novel and promising contribution.